# Understanding the trends in Reflected Solar Radiation: A Latitude- and month-based Perspective

Ruixue Li[1], Bida Jian[1], Jiming Li[1], Deyu Wen[1], Lijie Zhang[1], Yang Wang[1], Yuan Wang[1]

[1]Key Laboratory for Semi-Arid Climate Change of the Ministry of Education, College of Atmospheric Sciences, Lanzhou University, Lanzhou 730000, China

*Correspondence to:* Bida Jian (jianbd16@lzu.edu.cn) and Jiming Li (lijiming@lzu.edu.cn)

**Abstract.** Averaging reflected solar radiation (RSR) over the whole year/hemisphere may mask the inter-month/region-specific signals, limiting the investigation of spatiotemporal mechanisms and hemispheric symmetry projections. This drives us to explain RSR characteristics from latitude- and month-based perspectives. The study also explores whether longer-record radiation datasets can exhibit hemispheric symmetry of RSR to understand its temporal changes. Statistics indicate that the largest trends in decreasing RSR in Northern and Southern Hemispheres (NH and SH) occur in mid-spring and are dominated by (clear-sky atmospheric and cloud components), and cloud component only, respectively. The interannual negative trend in NH RSR mainly derives from 30°-50°N latitude zones, attributed to decrease in clear-sky atmospheric component caused by reduced anthropogenic sulphate emissions and spring/summer dust frequencies, and reduced cloud fraction caused by increased sea surface temperature and unstable marine boundary layer, leading to reduced cloud component. In SH, the significant RSR decreasing trend is widespread in 0°-50°S latitude zones, closely related to the decrease in cloud component caused by the decrease in cloud cover over the tropical western Pacific and Southern Ocean, partially compensated by the increase in clear-sky atmospheric component. A new data evaluation system and uncertainty analysis reveal that only AVHRR outperforms in exhibiting CERES's hemispheric RSR differences due to offsetting biases among different components, and achieves hemispheric RSR symmetry criteria within its uncertainty, making it suitable for studying long-term RSR hemispheric symmetry changes. Furthermore, ISCCP agrees well with CERES regarding hemispheric cloud component asymmetry and can help to study corresponding long-term changes and mechanisms.

## 1    Introduction

Planetary albedo (PA) refers to the fraction of incident solar radiation that is reflected back into

space by the Earth's atmosphere, clouds, and surface. It plays a crucial role in regulating the Earth's energy budget and global climate change (Wielicki et al., 2005; Stephens et al., 2015) by determining the amount of solar energy absorbed and distributed by the Earth-atmosphere system (Fu et al., 2000; Stephens et al., 2015). Studies have shown that a 5% change in PA can lead to an average global temperature change of approximately 1K (North et al., 1981), while a 3% change in PA can have a radiative forcing effect equivalent to doubling the amount of carbon dioxide in the atmosphere (Wielicki et al., 2005; Bender et al., 2006). Even small variations in PA could be sufficient for the development of Quaternary glaciations (Budyko, 1969). Therefore, it is essential to quantify the basic statistical properties of PA, and clarify the major principles governing its spatial-temporal changes and long-term trends at various scales, including annual, global, and even finer spatial-temporal scales (e.g., regional and monthly scales).

Nowadays, satellite data and model simulations have been widely used to investigate the climatology (Datseris and Stevens, 2021; Jönsson and Bender, 2022), spatial and temporal distribution characteristics (Loeb et al., 2007; Pang et al., 2022), and long-term trends of PA (Diamond et al., 2022; Stephens et al., 2022; Xiao et al., 2023), as well as the contributions of different components (e.g., cloud, clear-sky atmosphere, and surface) to PA (Stephens et al., 2015; Jönsson and Bender, 2022). Long-term satellite records have indicated that the current PA maintains a relatively stable value of approximately 0.29 (Bender et al., 2006). Surprisingly, the annual mean reflected solar radiation (RSR) in the Northern Hemisphere (NH) and Southern Hemisphere (SH) is almost same within measurement uncertainty, which is referred to as hemispheric symmetry (Loeb et al., 2009; Voigt et al., 2013; Stephens et al., 2015; Jönsson and Bender, 2022). However, although satellite observations have demonstrated the symmetry of hemispheric RSR on inter-annual scales, state-of-the-art models still struggle to reproduce this essential feature due to inadequate representation of the underlying physical mechanisms for RSR variation, particularly the poor modeling of compensatory effects of asymmetric clouds (Voigt et al., 2013; Stephens et al., 2015; Jönsson and Bender, 2022). As a result, mean hemispheric asymmetries persist in all-sky reflections from the Coupled Model Intercomparison Project (CMIP) phase 3 to phase 6, with considerable spread among the General Circulation Models (GCMs) within each CMIP phase (Crueger et al., 2023). Additionally, models also fail to capture the observed decreasing trend in RSR in both hemispheres. These limitations may stem from the inability of models to accurately simulate the

components of RSR and their respective contributions to the hemispheric symmetry of RSR. In fact, the annual mean RSR at the hemispheric scale is consisted of the RSR at finer spatial and temporal scales (such as regional and monthly scales). However, those signals of latitudinal and monthly variations are easily masked by studies at hemispheric or annual scales. It means that if models cannot accurately simulate the contribution of each component to hemispheric RSR at finer temporal and spatial scales, it will be bound to limit our ability in identifying potential regional maintenance or compensation mechanisms for hemispheric symmetry in RSR. Finally, above biases in RSR at finer temporal and spatial scales will exacerbate the uncertainties in model simulations of RSR at annual and hemispheric scales.

Indeed, decomposing the hemispheric annual RSR to finer spatial and temporal scales can help to identify the regional-scale influence and maintenance mechanism for hemispheric symmetry of RSR and further improve the model simulation of the radiative fluxes. Previous numerous studies have already demonstrated the importance of the regional compensation and influencing mechanism maintaining the hemispheric symmetry. For example, the Intertropical Convergence Zone (ITCZ) plays an important role in regulating cloudiness in the 10°S-10°N region, with its location and intensity varying seasonally (Waliser and Gautier, 1993; Hu et al., 2007). Based on the hemispheric-scale model simulations, early study conjectured that the ITCZ is the important compensating mechanism for the hemispheric symmetry of RSR by shifting it towards the darker surface hemisphere (Voigt et al., 2014). However, tropical clouds may not be the primary factor compensating for the hemispheric asymmetry of RSR, because the NH not only has the higher clear-sky albedo, but also the maximum tropical cloudiness (Jönsson and Bender, 2023). Nevertheless, based on finer temporal scales (such as monthly-scale) studies, it was found that variations in tropical clouds, especially those associated with the nonneutral phases of El Niño–Southern Oscillation (ENSO), are critical in regulating the asymmetry of hemispheric RSR (Jönsson and Bender, 2022). This suggests the importance of examining mechanisms influencing and maintaining hemispheric symmetry on finer spatial and temporal scales. Furthermore, extra-tropical cloudiness, particularly in the SH, has been highlighted as an important factor in maintaining the symmetry of the annual mean hemispheric albedo (Datseris and Stevens, 2021; Rugenstein and Hakuba, 2023). In addition, recent studies have emphasized the impact of the distinct land-sea distribution between hemispheres, which leads to enhanced baroclinic activities at mid-latitudes in the SH, resulting in an increase in baroclinic synoptic systems (Hadas et al., 2023). This activity results in intensified storm tracks, increased cloud

cover, and higher cloud albedo in the extratropical regions of the SH (Datseris and Stevens, 2021). These clouds effectively compensate for the asymmetry in clear-sky albedo between the NH and SH. The baroclinic activity at mid-latitudes exhibits a distinct seasonal cycle, with winter storm tracks in the NH being almost three times stronger than summer storm tracks (Hadas et al., 2023), and seasonal meridional shifts occurring in the SH (Verlinden et al., 2011). Besides, regional volcanic eruptions and forest fires also highly affect local atmospheric transmissivity and underlying surface albedo, even affect the albedo of polar snow cover remotely (Cole-Dai, 2010; Pu et al., 2021). These events usually occur in certain regions and forest fires occur typically during the summer and autumn (Fan et al., 2023), but they have important impacts on the interannual hemispheric symmetry of RSR.

In particular, note that the contributions of different latitudinal zones to hemispheric RSR are not independent of each other. Variations in the contributions of different latitudinal zones can offset or amplify each other, resulting in an energy balance or imbalance between the two hemispheres (hemispheric symmetry or asymmetry). For example, anthropogenic emissions from Asia not only enhance the local clear-sky atmospheric component of RSR through direct aerosol effects but also increase aerosol optical thickness in the northwestern Pacific through long-range transport. The long-range aerosol transport can also affect clouds by elevating cloud condensation nuclei (CCN) levels through the indirect effects of aerosols, thereby increasing cloud droplet number concentration, liquid water content, and updraft velocity. And it can increase the amount of deep convective clouds and lead to suppressed coalescence and warm rain but efficient mixed-phase precipitation (Zhang et al., 2007; Wang et al., 2014). The increased deep convective clouds and changed cloud microphysical processes over the northwestern Pacific can strengthen the NH storm track in the Pacific Ocean via large-scale enhanced convection and precipitation, thereby increase the contribution of the cloud component (Zhang et al., 2007; Wang et al., 2014). However, most of these studies are based on specific regions or components. Systematic studies on the distribution and changes of RSR and its components at finer temporal and spatial scales have received far less attention. Therefore, a comprehensive analysis of the contributions of different components at different latitudes and their monthly variations would help to better understand the mechanism of hemispheric RSR symmetry and reduce uncertainties in model simulations of RSR.

Currently, satellite remote sensing products from the CERES mission, which are based on broad-

band measurements, are invaluable for studying the energy balance of the Earth-atmosphere system (including changes in RSR and hemispheric symmetry) and climate change (Loeb et al., 2018b). In fact, researchers are still debating whether the hemispheric symmetry of RSR is an incidental outcome or an inherent feature of the Earth-atmosphere system. Based on CERES observations, a recent study found a decreasing trend in RSR in both hemispheres, while the hemispheric differences in RSR have not significantly changed (Jönsson and Bender, 2022), indicating that the hemispheric symmetry remains robust. Rugenstein and Hakuba (2023) suggested that hemispheric symmetry is a characteristic of the current climate state and may be disrupted in future scenarios. However, because the CERES observational record is relatively limited (2000-present), we cannot determine how hemispheric symmetry changes over time. Therefore, there is an urgent need for us to use longer and more reliable radiation records to verify the symmetry feature and explore the potential maintenance or compensation mechanisms of RSR symmetry. In recent years, satellite radiometric products and reanalysis data with longer time coverage and finer spatial resolution have been released, and numerous assessments have been conducted by researchers (Cao et al., 2016; Schmeisser et al., 2018; Loeb et al., 2022). The Cloud_cci version 3 radiative flux dataset has been shown to be in good agreement with the CERES EBAF dataset at a global scale (Stengel et al., 2020). Zhao et al. (2022) systematically assessed the applicability and accuracy of the Cloud_cci radiative flux dataset over the Tibetan Plateau (TP) and found that although the Advanced Very High Resolution Radiometer (AVHRR) can better describe the spatial and temporal characteristics of top-of-atmosphere (TOA) radiative fluxes over the TP, it does not capture the long-term trend of cloud radiative effects well. Furthermore, the spatial and temporal distributions of global TOA reflected solar radiation from Modern-Era Retrospective Analysis for Research and Applications, version 2 (MERRA-2) (Gelaro et al., 2017) and the fifth generation ECMWF reanalysis (ERA5) (Hersbach et al., 2020) have been compared with those from CERES (Lim et al., 2021), revealing that ERA5 shows better agreement with CERES than MERRA-2 in terms of seasonal fluxes. However, most of these assessments focus on the spatial and temporal reproducibility of these data in terms of global or regional radiative flux, while their performance in terms of hemispheric symmetry remains unknown. To understand the mechanisms maintaining hemispheric symmetry of RSR on longer time scales, it is essential to systematically quantify the performance of long-term radiative flux products in describing interhemispheric differences in TOA RSR and its components at hemispheric and finer

temporal-spatial scales. Additionally, identifying deficiencies and gaps between the datasets can provide

a reference basis for improving algorithms and parameterizations of radiation.

To enhance future investigations into the potential maintenance mechanisms of hemispheric symmetry and to reduce uncertainties in model simulations, this study aims to use long-term satellite observations of radiative flux (e.g., CERES-EBAF ed4.2) to quantify the contributions of clear-sky atmospheric, surface, and cloud components to RSR at finer spatial-temporal scales (e.g., regional and monthly scales). Additionally, we aim to analyze the spatial-temporal variability characteristics of these contributions. Furthermore, we will comprehensively evaluate the performance of various satellite and reanalyzed radiation datasets (including Cloud_cci AVHRR PM v3, ISCCP-FH, MERRA-2, and ERA5) in exhibiting the hemispheric differences and symmetry of CERES observed RSR and its components at hemispheric and finer temporal-spatial scales. The paper is structured as follows: Section 2 describes the data and methods used in the study; Section 3 presents the overall characterization (including: average and variability of RSR at different spatial and temporal scales), as well as the systematic assessment of different radiation datasets; and finally, Section 4 provides the conclusions and discussion.

## 2    Datasets and Methodology

### 2.1    Datasets

#### 2.1.1 CERES-EBAF

The Terra and Aqua satellites of the NASA were launched into Earth orbit in 1999 and 2002, respectively. Here, we use the products from Clouds and the Earth's Radiant Energy System (CERES) instrument flying on both the Terra and Aqua satellites to provide the monthly mean radiative flux.

CERES provides satellite-based observations to measure the Earth's radiation budget and clouds (Wielicki et al., 1996; Loeb et al., 2018b). The CERES instrument is a scanning broadband radiometer that provide radiation data across three channels: the shortwave channel (0.3–5μm), the infrared window channel (8–12μm), and the total channel (0.3–200μm). The radiance received by the CERES instrument is first converted from digital counts to calibrated "filtered" radiances. This is then converted to unfiltered radiances to correct for imperfections in the spectral response of the instrument (Loeb et al., 2001), and then transformed into TOA instantaneous radiative fluxes using an empirical angular distribution model (Su et al., 2015). Instantaneous fluxes are converted to daily-averaged fluxes using sun-angle dependent

diurnal albedo models (Loeb et al., 2018b). Surface irradiances are independently calculated using aerosols, clouds, and thermodynamic properties derived from satellite observations and reanalysis products. These calculations are constrained by the TOA irradiance (Kato et al., 2013; Kato et al., 2018).

Following Stephens et al. (2015) and Jönsson and Bender (2022), the study chooses the TOA and surface shortwave (SW) radiative fluxes from the CERES Energy Balanced and Filled (EBAF) product to analyze the contributions of different components. The CERES EBAF product employs an objectively constrained algorithm (Loeb et al., 2009) that adjusts the TOA SW and longwave (LW) fluxes within their uncertainties to remove inconsistencies between the global mean net TOA fluxes and the heat storage in the Earth-atmosphere system (Johnson et al., 2016). We use CERES EBAF, edition 4.2 (Loeb et al., 2018b), for monthly mean radiative fluxes (incident solar radiation, upwelling SW radiation at TOA, and both upwelling and downwelling SW radiation at the surface) during all-sky and clear-sky conditions between March 2001 and February 2022 (21 years) on a 1°×1° resolution grid. Note that EBAF data prior to June 2002 are Terra records only. In order to minimize flux discontinuities between the Terra-only record and the Terra&Aqua record, the CERES EBAF Ed4.2 product applies regional climate adjustments to the Terra-only record.

### 2.1.2 ISCCP-FH

The International Satellite Cloud Climatology Project (ISCCP) aims to provide global cloud coverage and cloud radiation characteristics (Schiffer and Rossow, 1983). As part of the ISCCP project, the ISCCP-FH radiation product contains SW radiation fluxes at five levels from the surface to the TOA (surface-680hPa-440hPa-100hPa-TOA) under all-sky, clear-sky and overcast-sky conditions as well as the diffuse and direct SW fluxes at the surface. ISCCP-FH is not produced using direct instrumental observations, but rather the ISCCP H series of data products that are derived from geostationary and polar-orbiting satellites (Young et al., 2018), adopting a complete radiative transfer model developed from the GISS GCM ModelE. As a third-generation product, ISCCP-H has become more advanced and has other improvements in radiation quality control, calibration, cloud detection (especially high clouds, thin clouds and polar clouds), cloud and surface properties retrievals (Zhang et al., 2023). The ISCCP-FH product consists of five sub-products, of which the PRF (surface-to-TOA flux profile) sub-product can provide 34 years of global radiative flux data from July 1983 to June 2017 with a spatial resolution of up to 1° and a temporal resolution of 3 hours. In order to be consistent with CERES EBAF data, this

study uses the diurnal mean of monthly mean of 3-hour upward and downward SW radiative flux at the TOA and surface under all-sky and clear-sky conditions provided by the MPF (monthly average of PRF) sub-product.

### 2.1.3 AVHRR

The Cloud_cci project covers the cloud component of the European Space Agency (ESA) Climate Change Initiative (CCI) program and has generated a long-term and consistent cloud property dataset (Hollmann et al., 2013). The Cloud_cci dataset is based on the state-of-the-art retrieval system called "the Community Cloud retrieval for Climate" (CC4CL), which employs optimal estimation (OE) techniques and is applied to passive imaging sensors from current and past European and non-European satellite missions (Sus et al., 2018). The Cloud_cci AVHRR-PMv3 dataset, which contains comprehensive cloud and radiative flux properties globally from 1982 to 2016, is chosen for the comparison with CERES EBAF. These properties are retrieved from measurements obtained by the AVHRR instrument onboard the afternoon (PM) satellite of the US National Oceanic and Atmospheric Administration's (NOAA) Polar Operational Environmental Satellite (POES) mission (Stengel et al., 2020). To account for the diurnal cycle of the solar zenith angle, all samples of the SW flux are rescaled and averaged to represent a 24-hour average for each pixel. The monthly average value is then determined (More details can be found in ESA Cloud_cci Algorithm Theoretical Baseline Document v6.2). Note that the radiation broadband flux is determined using exported cloud characteristics combined with reanalysis data (Stengel et al., 2020). However, there are some differences in this product for the years 1994 and 2000 due to the unavailability of AVHRR data. Therefore, data from these years are not used in this study. We use the monthly mean global 0.5° grid data (Level-3C) from Cloud_cci, which includes TOA and surface upward and downward SW radiative fluxes under both all-sky and clear-sky conditions and interpolate this data to a 1° grid to keep consistency with CERES.

### 2.1.4 Reanalysis datasets

In this study, we select two state-of-the-art reanalysis data to evaluate their applicability in the study of hemispherical symmetry: MERRA-2 and ERA5 reanalysis datasets.

MERRA-2 is the latest atmospheric reanalysis of the modern satellite era produced by NASA's Global Modeling and Assimilation Office (GMAO) with version 5.12.4 of the Goddard Earth Observing

System (GEOS) atmospheric data assimilation system (Gelaro et al., 2017). It is the first long-term global reanalysis to assimilate space-based observations of aerosols and represent their interactions with other physical processes in the climate system. MERRA-2 can provide long-term radiative products with a spatial resolution of 0.5°×0.625° from 1980. Here, M2TMNXRAD (or tavgM_2d_rad_Nx) monthly mean radiative flux data, including the incident and net downward SW radiative fluxes at the TOA and the surface under all-sky and clear-sky conditions, are used for comparative assessment with CERES data.

ERA5 is the fifth-generation atmospheric reanalysis of the global climate from January 1940 to present by the European Centre for Medium-Range Weather Forecasts (ECMWF). ERA5 combines model data with observations from around the world to form a globally consistent dataset that replaces the previous ERA-Interim reanalysis. 4D-var data assimilation technique in the Integrated Forecasting System (IFS) Cycle 41r2 is used to ensure a significant improvement in prediction accuracy and computational efficiency (Jiang et al., 2019; Hersbach et al., 2020). It provides hourly estimates of a large number of atmospheric, land and oceanic climate variables with a spatial resolution of 0.25°×0.25° (Hersbach et al., 2020). The monthly average surface and TOA radiation budget products are used in this study.

In order to maintain data consistency, the monthly mean diurnal averaged radiative fluxes from MERRA-2 and ERA5 datasets are resampled to match the 1°×1° resolution of CERES.

Note that for a more accurate comparison with CERES EBAF, the other radiative flux data mentioned above (SW radiative flux from ISCCP-FH, AVHRR, ERA5, and MERRA-2) have been selected for their overlapping time period from March 2001 to February 2016.

## 2.2  Methodology

### 2.2.1 Decomposition of reflected solar radiation contribution

To investigate the main drivers of the RSR, we use the similar model as Stephens et al. (2015) to decompose the RSR into the contributions of the surface and atmospheric components. Assuming that surface and atmospheric reflection and absorption processes are isotropic, planetary albedo R is defined as:

$$R = \frac{F_{\text{TOA}}^{\uparrow}}{S} \tag{1}$$

Among them, the $F_{\text{TOA}}^{\uparrow}$ is reflected SW (upwelling) flux at the TOA, S is the solar incident (downwelling) flux. The transmittance T of the whole Earth-atmosphere system is defined as:

$$T = \frac{F_{\text{S}}^{\downarrow}}{S} \tag{2}$$

Where, $F_{\text{S}}^{\downarrow}$ is the downwelling SW radiation at the surface. The surface albedo $\alpha$ is calculated as follows:

$$\alpha = \frac{F_{\text{S}}^{\uparrow}}{F_{\text{S}}^{\downarrow}} \tag{3}$$

Where $F_{\text{S}}^{\uparrow}$ is the upwelling SW radiation at the surface. The term $F_{\text{S}}^{\downarrow}$ can be expressed as:

$$F_{\text{S}}^{\downarrow} = tS + rF_{\text{S}}^{\uparrow} \tag{4}$$

Here, r and t represent atmospheric intrinsic reflectivity (that is, PA purely contributed by the atmosphere) and atmospheric transmittance, respectively. The r and t are calculated separately, so absorption and forward scattering are included in t. $F_{\text{TOA}}^{\uparrow}$ can be represented as:

$$F_{\text{TOA}}^{\uparrow} = rS + tF_{\text{S}}^{\uparrow} \tag{5}$$

By combining the above equations, R and T can be expressed by r, t and $\alpha$:

$$R = r + \frac{\alpha t^2}{1 - r\alpha} \tag{6}$$

$$T = \frac{t}{1 - r\alpha} \tag{7}$$

According to the above equation, the values of r and t can be written:

$$r = R - t\alpha T \tag{8}$$

$$t = T\frac{1 - \alpha R}{1 - \alpha^2 T^2} \tag{9}$$

It can be seen that the planetary albedo R is composed of two parts: atmospheric contribution r and surface contribution $\frac{\alpha t^2}{1-r\alpha}$. These two parts are multiplied by the incident solar radiative flux S respectively, and the respective contribution values of the atmosphere and the surface to the RSR at TOA ($F_{\text{TOA}}^{\uparrow}$) can be obtained, namely $F_{\text{atm}}^{\uparrow}$ and $F_{\text{surf}}^{\uparrow}$ (unit: W m$^{-2}$).

$$F_{\text{atm}}^{\uparrow} \equiv Sr \tag{10}$$

$$F_{\text{surf}}^{\uparrow} \equiv S\frac{\alpha t^2}{1 - r\alpha} \tag{11}$$

Following Stephens et al. (2015) as well as Jönsson and Bender (2022), we further decompose the atmospheric component into clear-sky atmospheric and cloud contributions. The difference between the all-sky atmospheric contribution $F_{\text{atm}}^{\uparrow}$ and the clear-sky atmospheric contribution $F_{\text{atm,clear}}^{\uparrow}$ is

considered as the cloud contribution $F_{\text{cloud}}^{\uparrow}$. That is,

$$F_{\text{TOA}}^{\uparrow} = F_{\text{atm}}^{\uparrow} + F_{\text{surf}}^{\uparrow} = F_{\text{cloud}}^{\uparrow} + F_{\text{atm,clear}}^{\uparrow} + F_{\text{surf}}^{\uparrow} \tag{12}$$

$$F_{\text{cloud}}^{\uparrow} = F_{\text{atm}}^{\uparrow} - F_{\text{atm,clear}}^{\uparrow} \tag{13}$$

### 2.2.2 Regional mean and contribution rate

In calculating regional averages radiative flux, the study employs a geodesic weighting method consistent with the official CERES product. This method assumes Earth's oblate spheroid shape and takes into account the annual cycle of the Earth's declination angle and the sun-Earth distance (details about the method can be found in the website: "https://ceres.larc.nasa.gov/documents/GZWdata/ zone_weights.f"). The regional averaged TOA RSR $F_k$ is spatially aggregated using the following calculation formula:

$$F_k = \frac{\sum_{i=1}^{N_k} W_{ki} \cdot F_{ki}}{\sum_{i=1}^{N_k} W_{ki}} \tag{14}$$

Here, $N_k$ is the number of grid samples in region k, and $F_{ki}$ is the RSR flux corresponding to grid i in the region k. Moreover, $W_{ki}$ is the geodetic zonal weight for the grid i, which can be obtained from "https://ceres.larc.nasa.gov/documents/GZWdata/zone_weights_lou.txt". Regional averages for other variables are calculated according to the similar weighting equation.

In order to explore the contribution of different regions to the total hemispheric RSR, the global latitude is divided into 18 latitude zones in the unit of 10°, that is, 90°N-80°N, 80°N-70°N, …, 70°S-80°S, 80°S-90°S. For example, the rate of the cloud component contribution $C_{\text{cloud}}$ of each latitude zone to its hemispheric RSR can be calculated by the following formula:

$$C_{\text{cloud}} = \left( \frac{total\_latzone\_cloud}{total\_hem\_R} \right) \times 100\% \tag{15}$$

Where $total\_latzone\_cloud$ refers to the sum of the latitude-weighted RSR of cloud component from all grids in the given latitude zone, $total\_hem\_R$ is the sum of latitude-weighted total RSR from all grids in the hemisphere in which the latitude zone is located. The contribution of surface and clear-sky atmospheric components to hemispheric RSR in different latitudinal zones can be derived by the similar method.

### 2.2.3 Time average

For the average contribution over time, we consider March to the following February as a complete

year. Following the CERES EBAF Ed4.1 Data Quality Summary (2020), the monthly average data is weighted by the number of days in each month to obtain the annual average data (Wielicki et al., 1996; Loeb et al., 2009; Rugenstein and Hakuba, 2023). For example, the annual average value of TOA RSR in a certain year is:

$$F_{Year} = \sum_{i=1}^{i=12} \frac{DAY_{mon}(i)}{DAY_{year}} F_{mon}(i) \tag{16}$$

where $DAY_{year}$ is the total number of days in the given year, $DAY_{mon}(i)$ is the number of days in the current month, and $F_{mon}(i)$ is the monthly averaged RSR. The annual average values of all variables are also obtained by this method.

**2.2.4 CCHZ-DISO data evaluation system**

To find out whether other radiation datasets can exhibit the similar hemispheric symmetry of RSR, the CCHZ-DISO data evaluation system is also used. This method uses the Euclidean Distance between indices of simulation and observation (DISO) to evaluate the combined quality or overall performance of data from different models (Hu et al., 2019; Zhou et al., 2021; Hu et al., 2022). DISO has the advantage of quantifying the combined accuracy of different models compared to Taylor diagram (Kalmár et al., 2021). Moreover, the statistical indicators chosen for the Taylor diagram are fixed, whereas those in DISO can be taken and discarded according to the needs of the study (Hu et al., 2022). In particular, Taylor diagrams only provide statistical metrics on two-dimensional plots, DISO not only provides distances in three-dimensional space to quantify the comprehensive performance of a simulation model, but also allows a single statistical metric to capture different aspects of model performance (Hu et al., 2019).

In this paper, CERES-EBAF during March 2001-February 2016 is taken as the observed dataset, while AVHRR, ISCCP, MERRE-2, ERA5 are considered as the model datasets. For the observed time series and the model-simulated time series, their correlation coefficient (CC), absolute error (AE), and root mean square error (RMSE) are obtained from Eqs. (17-19), respectively.

$$CC = \frac{\sum_{k=1}^{n}(a_k - \bar{a})(b_k - \bar{b})}{\sqrt{\sum_{k=1}^{n}(a_k - \bar{a})^2}\sqrt{\sum_{k=1}^{n}(b_k - \bar{b})^2}} \tag{17}$$

$$AE = \frac{1}{n}|\sum_{k=1}^{n}(b_k - a_k)| \tag{18}$$

$$\text{RMSE} = \sqrt{\frac{1}{n}\sum_{k=1}^{n}(b_k - a_k)^2} \tag{19}$$

The CCHZ-DISO 3D evaluation system is then constructed using NCC, NAE and NRMSE, which

are normalized CC, AE and RMSE, respectively. Please note that the metrics are normalized to be

between 0 and 1, using the normalization formula following Chen et al. (2024) as:

$$NS_a = \frac{S_a - \min(S)}{\max(S) - \min(S)} \tag{20}$$

Where S indicates the metric (CC, AE, and RMSE). Here, a=0, 1, …, m, "0" indicates the observed

data, and m is the total number of model data used for comparison.

$$DISO_i^{xj} = \sqrt{(NCC_i - NCC_0)^2 + (NAE_i - NAE_0)^2 + (NRMSE_i - NRMSE_0)^2} \tag{21}$$

Where $i$ and $xj$ represent the $i$th model and $j$th variable. The subscript "0" in Eq. 21 represents

statistical parameters of variable $xj$ from observation data (here refers to CERES EBAF). A

smaller/larger $DISO_i^{xj}$ values indicates better/worse performance of model $i$ in simulating variable $xj$.

## 3    Results

### 3.1    Temporal variation of RSR components in different latitudinal zones

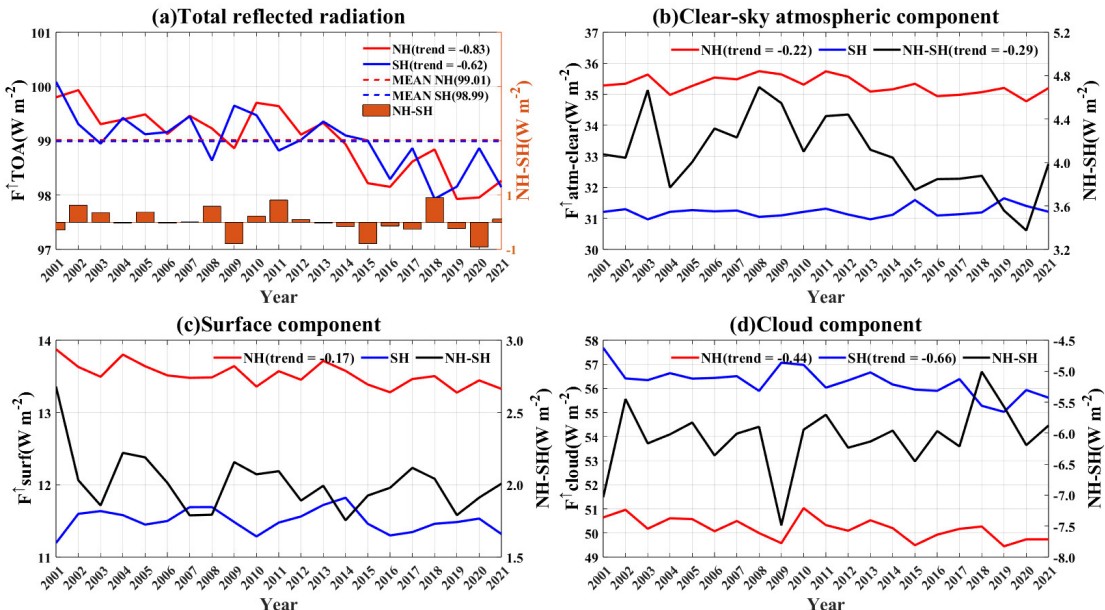

**Figure 1: The interannual mean time series of (a) total reflected solar radiation at the TOA and its (b)clear-sky atmospheric, (c)surface, and (d) cloud components in NH and SH (the left axis), as well as the difference between NH and SH (the right axis) from 2001 to 2021.Note that the scales of the two y-axes are not the same. The red line is for the NH, the blue line is for the SH, the orange bars and black line are for hemispheric difference (NH-SH), and the dashed line is the 21-year average values. The trends marked passes the 95% significance test in units of W m$^{-2}$ decade$^{-1}$.**

Firstly, we examine the general characteristics of reflected radiation in the NH and SH on an annual average scale. Figure 1 illustrates the interannual variability of RSR at the TOA and its three components in the NH and SH during the period of 2001-2021, based on CERES EBAF data. The RSR in both hemispheres shows symmetry in term of multi-year averages (21-year average difference: 0.02 W m$^{-2}$) and the long-term trends. Both hemispheres exhibit a consistent decreasing trend in total RSR (Trend_NH=-0.83 W m$^{-2}$ decade$^{-1}$; Trend_SH=-0.62 W m$^{-2}$ decade$^{-1}$), indicating simultaneous darkening of both hemispheres as observed from space, with the NH darkening at a faster rate. To investigate whether these trends in RSR are linked to changes in incident solar radiation, we also present the interannual variations of incident solar radiation and PA (Fig. S1). The results indicate that the interannual variations of incident solar radiation at TOA in both hemispheres do not exhibit a significant trend, with the hemispheric difference following a stable multi-year cycle. However, PA in both hemispheres shows a consistent decreasing trend (Trend_NH=-2.4×10$^{-3}$ decade$^{-1}$; Trend_SH=-1.8×10$^{-3}$ decade$^{-1}$), suggesting a decrease in RSR by the Earth as a whole and an increase in absorbed solar radiation. However, the same response in both hemispheres is driven by different component changes. The darkening of the SH can be primarily attributed to a decrease in RSR from the cloud component (-0.66 W m$^{-2}$ decade$^{-1}$) (Fig. 1d). In contrast, the RSR by three components in the NH all show a decreasing trend, with the cloud component exhibiting the largest decrease (-0.44 W m$^{-2}$ decade$^{-1}$), followed by the clear-sky atmospheric component (-0.22 W m$^{-2}$ decade$^{-1}$), and the smallest decrease is for the surface component (-0.17 W m$^{-2}$ decade$^{-1}$). Moreover, the hemispheric asymmetry (NH-SH) of the clear-sky atmospheric component is decreasing (-0.29 W m$^{-2}$ decade$^{-1}$) around 2008 year, which is mainly influenced by the declining reflection of the clear-sky atmosphere in the NH due to the reduced scattering of aerosol particles (Loeb et al., 2021a; Stephens et al., 2022; Diamond et al., 2022).

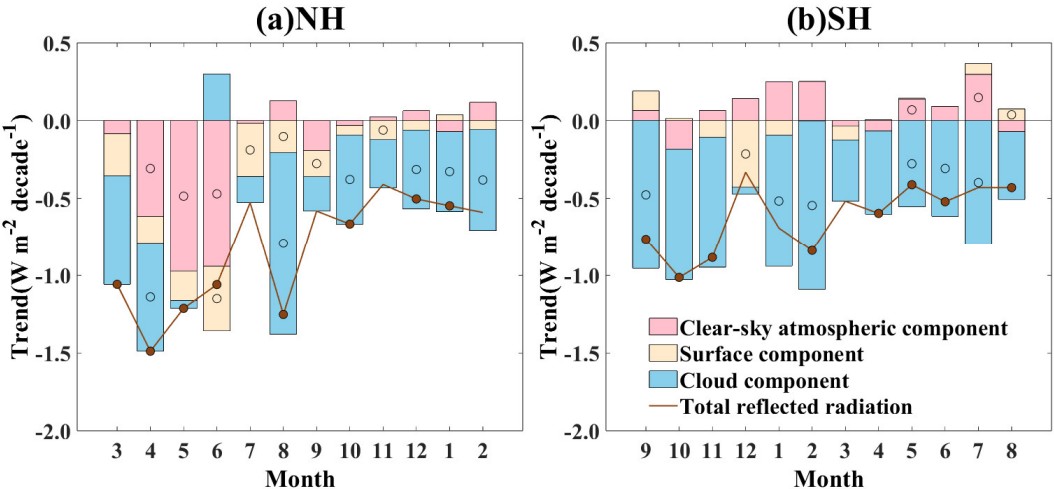

**Figure 2: The hemispheric averaged trends in reflected solar radiation and its components in the (a) NH and (b) SH for different month from 2001-2021. Pink, yellow and blue bars indicate trends in the clear-sky atmospheric component, surface component and cloud component, respectively. The brown line indicates the trend of total reflected solar radiation. Dots of different colours indicate that the hemispheric averaged trend of the corresponding variable is significant at the 95% confidence level.**

The analysis presented above is based on the results of annual average RSR. Note that the symmetry of RSR between hemispheres is a characteristic observed at interannual scales. However, certain natural and human activities (e.g., the Pinatubo eruption, Australian bushfires, societal response to the COVID-19 pandemic) that strongly influence albedo or compensate for hemispheric asymmetry are seasonal or even occur only in specific months of the year (Minnis et al.,1993; Hirsch and Koren, 2021; Diamond et al., 2022). They can generate large perturbations on interannual scales due to strong signals in specific seasons. To further clarify the variations of these mechanistic signals by resolving RSR and its components at finer temporal scale (e.g., monthly), Figure 2 resolves the long-term trends in RSR for both hemispheres into different months. The results indicate that the significant decreasing trends in hemispheric RSR for both hemispheres are generally observed throughout the year, with an obvious reduction from spring to winter. This is related to seasonal changes in trends with different components. Considering the potential impact of the annual cycle of incident solar radiation, we have also provided the hemispheric average monthly trends of TOA incident solar radiation (Fig. S2) and PA (Fig. S3). The results show that there are no significant long-term trends in the incident solar radiation among all months. However, after removing the effects of seasonal variations in incident solar radiation, the monthly trends in PA for both hemispheres exhibit distinct differences from the total RSR. Under these conditions, the decreasing trends in PA during winter are comparable to those observed during summer in both

hemispheres. This suggests that the weaker decreasing trend of RSR in autumn and winter compared to that in spring and summer may be affected by the reduced incident solar radiation during these seasons. In addition, there is no significant trend in the RSR of the NH for July and the SH for December. This may be due to the fact that the decreasing trends observed in different months are regulated by different components at different latitudes, thus not showing consistent changes.

In the NH, the decreasing trends of RSR are highest in the months of March to June and August, being more than twice as large as the trends in winter months. The peak value of the decreasing trend occurs in April, which is influenced by both the clear-sky atmospheric and cloud components. The trend from April to June is primarily driven by the clear-sky atmospheric component. Here, we further decompose the results of the monthly trend into different latitude zones (Fig. S4, S5), the statistical results show that the significant decreasing trend of the clear-sky atmospheric component in the NH during April-June is mainly contributed by the mid-latitude regions (30°N-60°N). The vital dust belt is located in these regions, serving as the major emission source of dust, typically peaking in spring and early summer (Yang et al., 2022). However, due to reduced local wind speeds and increased soil moisture, dust activity frequencies in regions such as West Asia, and Central Asia have experienced varying degrees of decline (Shao et al., 2013; Shi et al., 2021; Zhou et al., 2023). Particularly, the frequency of dust storms in China has notably decreased due to increased vegetation cover (Zhao et al., 2018; Jiao et al., 2021). Moreover, in regions with concentrated industrial and anthropogenic aerosol emissions, such as Europe, eastern and central China and North America, effective emission reduction policies have led to a decrease in polluting sulfate aerosols (Zhao et al., 2017; Li et al., 2020; Tao et al., 2020; Yu et al., 2020; Gui et al., 2021; Cui et al., 2022; Tang et al., 2022), weakening the contribution of the clear-sky atmospheric component in RSR. In most months (especially in August, October, December, and January) except the spring, the decreasing trend of RSR in the NH is primarily dominated by the cloud component. The decreasing trend of cloud component reaches its maximum in August, and is mainly influenced by the regions between 50°N-60°N and 0°-10°N. For the 50°N-60°N regions, the low cloud cover over northeast Pacific has decreased markedly over the last 20 years, due to the weakening temperature inversion intensity and increasing sea surface temperature (SST), which has reduced the cloud component of RSR in this region (Andersen et al., 2022). At 0-10°N, the decreasing trend in RSR is particularly strong over the tropical western Pacific. This is due to the increase in SST, which reduces the stability of the marine

boundary layer (MBL), leading to MBL deepening and decoupling between cloud cover and surface moisture supply, thus reducing the cloud cover and corresponding cloud component of RSR (Loeb et al., 2018a). Compared to the other components, the surface component of RSR does not dominate the decreasing trend of NH in a specific month. It decreases most rapidly in June, followed by July, which is primarily located at the region between 70°N-80°N. This decrease may be related to the advancement and lengthening melting period of Arctic ice due to the Arctic amplification effect, which can affect changes in surface component of RSR (Noël et al., 2015; Wang et al., 2018; Mika et al., 2022).

In the SH, the cloud component dominates the decreasing trend of RSR for all months except December. This dominant role is mainly contributed by the latitudinal zones from equator to 60°S, although the trends of cloud component in these latitudinal zones may not be significant on a single month. This may be partly attributed to decreasing cloud cover in specific regions, such as the tropics and the Southern Ocean. On the one hand, the low cloud cover over tropics has decreased due to the increasing SST. On the other hand, multi-source satellite cloud climatological data consistently show a significant decreasing trend in total cloud cover over the Southern Ocean (Devasthale and Karlsson, 2023). The maximum value of the RSR decreasing trend occurs in October, while the cloud component of RSR decreases fastest in February. In December, the trend in the SH is dominated by the surface component in the region of 60°S-70°S (see Figure S5), where it is covered with extensive ice and snow coverage. Under the background of global warming, ice and snow are melting rapidly, resulting in remarkable seasonal changes in ice and snow cover.

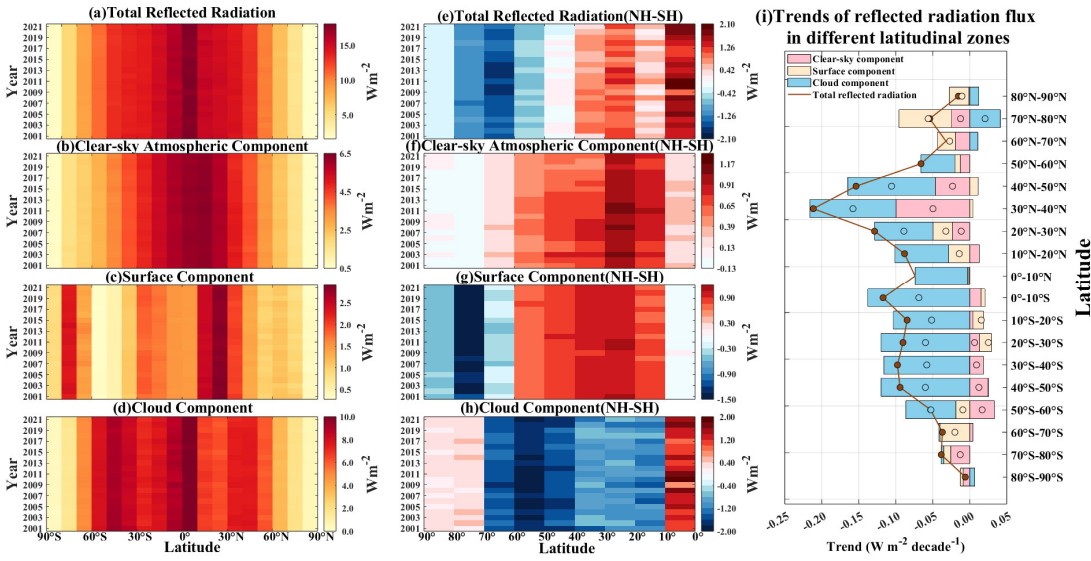

**Figure 3: Annual averaged time series of (a) total RSR and its (b) clear-sky atmospheric component, (c) surface component, and (d) cloud component at different latitudinal zones, along with (e-h) the**

Large-scale systems or certain compensatory mechanisms that may affect the hemispheric symmetry of RSR do not directly act on a hemispheric scale. Instead, they can compensate for hemispheric energy imbalances by affecting local or regional climates. For example, baroclinic activity, although occurring mainly at mid-latitudes, has a great impact on cloud albedo, thereby strongly impacting global albedo (Hadas et al., 2023). While larger regional anomalies in RSR may offset each other when spatially and temporally averaged to calculate global RSR and its interannual variations, these anomalies play a crucial role in regional radiation budgets, subsequent climate change, and the identification of mechanisms that maintain or compensate for hemispheric symmetry of RSR. Therefore, to further deepen the understanding of the regional RSR changes and provide a reference for mechanism research, we divide the globe into 18 latitudinal zones in 10° increments. Figure 3a-h show the time series of latitudinal averaged RSR and interhemispheric differences of RSR and their components at different latitudinal zones, and Figure 3i shows the interannual trends of RSR and its components at different latitudinal zones. Note that the RSR and their trends for different latitudinal zones are area-weighted based on Eq. (14) for comparison. In general, the total RSR in both hemispheres decreases from the equator towards the poles, while the zonal-averaged magnitude of their components of RSR varies. In the SH, the zonal distribution of clear-sky atmospheric components is similar to that of the RSR. In the NH, the extreme values of clear-sky atmospheric components of RSR occur at 10°N-20°N, where there is a large amount of dust aerosols from the Sahara Desert. The RSR peak by surface components are located at 70°-80° in the SH and 20°-30° in the NH, respectively, due to the high ice and snow albedo and high surface albedo caused by bare ground. The cloud component reflects the most radiation at 40° S-50°S in the SH and at 0°-10°N in the NH, since these regions are where the storm tracks of Southern Ocean (Datseris and Stevens, 2021) and the annual average position of ITCZ (Gruber, 1972) are located, respectively.

For the hemispheric differences (Fig. 3e-h), it is shown that more energy is reflected from the 0°-40° latitude zones in the NH compared to the corresponding latitude zones in the SH. However, this imbalance is compensated by more reflection from the SH in the 50°-90° latitude zones. The higher RSR

from the 0°-40° latitude zones in the NH stems from the higher cloud component from the equator to 10° and the combined effect of clear-sky atmospheric and surface components in the 10°-40°. In contrast, the strength of the SH at middle and high latitudes is derived from the surface component from 60°-90° and the cloud component from 40°-70°. At 40°-50°, more reflected radiation from cloud component in the SH offset the more radiation from clear-sky atmospheric and surface components in the NH. Regarding the clear-sky atmospheric component, the NH as a whole is slightly higher than the SH (except in the polar regions), possibly due to the large amount of dust aerosols in the NH tropics and subtropics, as well as more sulfate pollution in the mid-latitudes (Diamond et al., 2022). Notably, the difference in clear-sky atmospheric components between the two hemispheres is greatest at 20°-30°, influenced by the combined effect of more dust and sulfate aerosols in the NH. There are significant hemispheric differences in surface component, with the NH exhibiting larger RSR from surface component concentrated in the 10°-60° latitude range because of the larger land area in the NH. At the high latitudes of 70°-80°, the SH shows larger surface reflections due to higher snow and ice cover in the near polar regions. Between 50°-60°, cloud components in the SH reflect more solar radiation and reach maximum hemispheric differences. This is attributed to the higher subtropical cloudiness and cloud albedo at mid-latitudes (Bender et al., 2017). The more radiation from NH clouds near the equator may be due to the persistent presence of the ITCZ north of the equator in the eastern Pacific and Atlantic. This observation suggests that the SH heavily relies on extratropical clouds to compensate for clear-sky hemispheric asymmetries, which is consistent with previous studies (Datseris and Stevens, 2021; Blanco et al., 2023; Hadas et al., 2023; Rugenstein and Hakuba, 2023). Based on the above analyses, we can find that the RSR and its components in the corresponding latitude zones of the two hemispheres are asymmetric. It is the offsetting of the differences in the different components across the latitudinal zones that leads to the minimal hemispheric differences in total RSR -- the cloud component in the mid-latitudes and the surface component in the high latitudes of the SH offset the clear-sky reflectance in the mid-low latitudes of the NH.

In addition, to clarity the variations and hemispheric differences of RSR at finer temporal scale, we further analyze the annual cycle of hemispheric differences of RSR across different latitudinal zones. Figure S6-S9 illustrate the annual cycle of RSR and its components in different latitudinal zones and their interhemispheric differences. It can be seen that the hemispheric differences of RSR in different

latitudinal zones present obvious monthly variations, with the peak values in summer and winter. At middle-high latitudes, the annual cycles of the hemispheric differences of RSR and its components is relatively consistent with that of incident solar radiation (Fig. S10). However, the surface components in the 40°-60°N latitudinal zones exhibit enhanced reflectivity and interhemispheric differences in spring (Fig. S8), possibly influenced by surface albedo (Fig. S11). The annual cycle of hemispheric RSR differences is dominated by the cloud component at mid-low latitude and the surface component at high-latitude.

Furthermore, statistical results indicate that the decadal trends of RSR at different latitudinal zones are highly significant (Fig. 3i). It is clear that the hemispheric decreasing trend of RSR is the cumulative result of decreasing trends of RSR across all latitude zones. Fig. S12a-c presents the global distribution of the trends in three components of RSR, which help to identify the key areas and factors influencing the trends. From Fig. 3i, it can be observed that the RSR trends in the NH for different latitude zones below 60°N are widely different, whereas the trends in the SH for different latitude zones below 60°S are relatively homogeneous, mainly due to the difference in their dominant components. Most of the downward trends in the NH come from 20°-50°, with the strongest trend coming from 30°-40°, dominated by significant decreases in cloud and clear-sky atmospheric components. Decreasing trends in cloud component are mainly observed over the Northeast Pacific and North Atlantic near North America (Fig. S12c). The decreasing trend in cloud component over the Northeast Pacific may be associated with a shift in the Pacific Decadal Oscillation (PDO) phase from negative to positive, which leads to warmer SSTs in parts of the eastern Pacific, thus reducing low cloud cover and RSR (Loeb et al., 2018a; Loeb et al., 2020; Andersen et al., 2022). And the reduction in the North Atlantic cloud component may be related to a reduction in the optical thickness of low clouds due to a reduction in AOD (Park et al., 2024). The significant decreasing trends for 20°N-50°N in the clear-sky atmospheric component occurs in Europe, central China, the eastern seas of China and the eastern United States (Fig. S12a), which is consistent with previous studies and related to the reduced aerosol particle scattering (Loeb et al., 2021a; Raghuraman et al., 2021; Quaas et al., 2022; Stephens et al., 2022). At 70°N-80°N, the decreasing trend in total RSR is dominated by the surface component, accompanied by a significant decrease in the clear-sky atmospheric component and partially compensated by an increase in the cloud component. The strong downward trend of the surface component can be observed along the northern

coast of the Asian and European continents and over the Arctic Ocean (Fig. S12b), which is inseparable from the decrease in albedo caused by the strong retreat of sea ice.

From the equator to 60°S, there are significant decreasing trends in cloud components, which dominate the trends in RSR (Fig. 3i). The extreme value of the trends in total RSR of SH occurs at 0°-10°S due to the significant reduction in cloud components over the tropical western Pacific (Fig. S12c). From 20°S-60°S, the trends in clear-sky atmospheric component even exhibit significant positive values, especially at 50°S-60°S. The increasing trend of clear-sky atmospheric component in low-latitude zones

of SH is primarily observed over Chile and the South Tropical Pacific (Fig. S12a). This trend in the former region stems mainly from the increasing secondary aerosol loading (Miinalainen et al., 2021), while the trend in the latter region may be remotely influenced by biomass burning in South-East Asia and South America (Li et al., 2021). In addition, studies have shown that large amounts of dust and smoke from the 2019-2020 forest fires in Australia greatly affect the aerosol loading over the South Pacific

(Yang et al., 2021). At mid and high latitudes, the clear-sky atmospheric components are generally increasing over the Southern Ocean, which may be related to the change of aerosol loading. Based on model simulations, Bhatti et al. (2022) found that the depletion of stratospheric ozone can alter the westerly jet and affect wind-driven aerosol fluxes, hence increasing the aerosol loading over the Southern Ocean, which includes sea salt aerosols and phytoplankton-produced sulfate aerosols. In addition, the

reduction in clouds may also contribute to the increase in clear-sky atmospheric component. This is because cloud cover may mask some reflection of clear-sky components such as aerosols below the clouds (Qu and Hall, 2005; Donohoe and Battisti, 2011; Voigt et al., 2014; Stephens et al., 2015). Naturally, the decrease in cloud cover may reveal a portion of clear-sky atmospheric component.

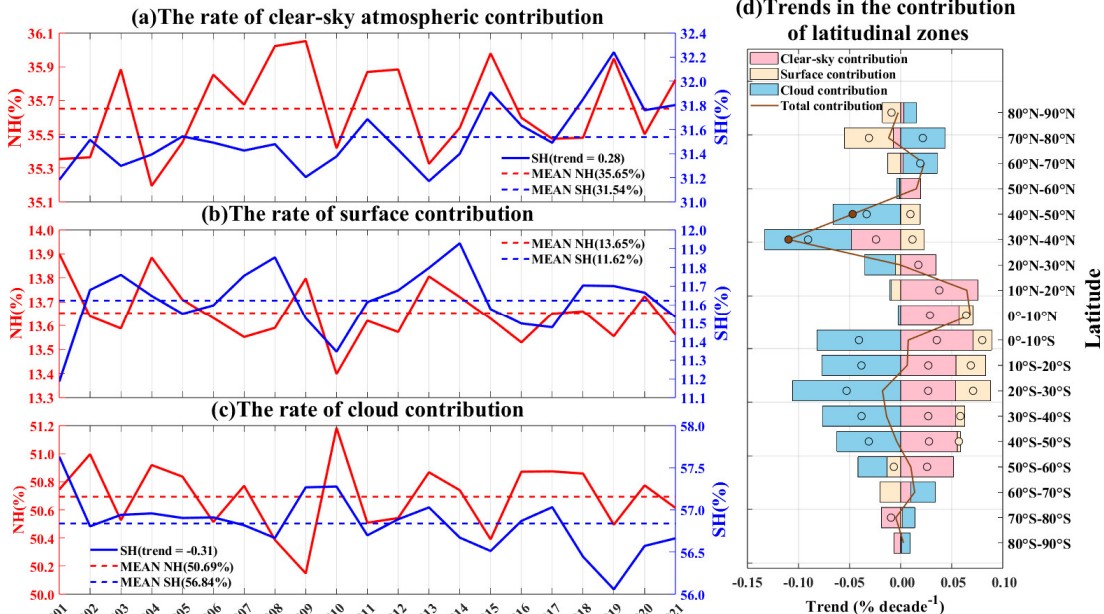

**Figure 4: Interannual mean time series of the contribution rate for (a) the clear-sky atmospheric component, (b) the surface component, and (c) the cloud component to the total reflected solar radiation at the TOA in the NH (the left axis) and SH (the right axis) from 2001-2021; note that the scales of the two axes are not the same. The red line is for the NH, the blue line is for the SH, and the red/blue dashed lines are 21-year averaged values of NH/SH. The trends marked in the upper right corner passes the 95% significance test in units of % 570 decade$^{-1}$. (d)The zonal mean trends in the contribution rate of different latitudinal zones to hemispheric total reflected solar radiation from 2001-2021. Pink, yellow and blue bars indicate trends in the clear-sky atmospheric contribution, surface contribution and cloud contribution, respectively. The brown line indicates the trend of total reflected solar radiation contribution. Dots of different colours indicate that the zonal mean trend of the corresponding variable at the given latitude zone is significant at the 95% confidence level.**

The analysis above is all based on RSR and its components at different latitudinal zones, which can directly show the variation of their reflected ability to solar radiation. However, they cannot reflect changes and adjustments in the contribution of different components to the total RSR. So this study further quantifies the contribution rates of different components to the RSR (Fig. 4a-c) and the contribution rates of different latitudinal zones to hemispheric RSR based on Eq. (15) (Fig. 4d). There 580    are clear hemispheric asymmetries in the contributions of the three components to the hemispheric RSR, which indicates that the relative importance of the three components varies in different hemispheres. For both hemispheres, the cloud component contributes the most to the RSR, accounting for over 50%, followed by the clear-sky atmospheric component, while the surface component contributes the least. The cloud contribution rate in the SH is approximately 6.15% higher than that in the NH, which can be 585    attributed to more and brighter clouds in the SH (Stephens et al., 2015; Datseris and Stevens, 2021; Diamond et al., 2022; Jönsson and Bender, 2023). The clear-sky atmospheric contribution rate in the NH

is 4.11% higher than that of the SH, possibly due to greater anthropogenic aerosol emissions resulting from human activities in the NH (Diamond et al., 2022; Jönsson and Bender, 2022). Although all three components of RSR in the NH show significant decreasing trends, there is no significant trend in the proportion of their contributions. This means that there is no major adjustment in the radiation budget for the NH. The clear-sky atmospheric contribution rate in the SH shows an increasing trend of 0.28% per decade, which may be regulated by a decreasing trend of -0.31% per decade in the cloud component contribution (Fig. 1 and Fig. 4). Compared to the SH, the NH exhibits a 2.03% higher surface contribution rate. Although the NH has a larger land distribution, the higher ice albedo in Antarctica partially compensates for the lack of land area in the SH (Fig. S11), resulting in a minor difference in surface contribution between the hemispheres (Diamond et al., 2022).

The spatial distributions of the contribution rates of the three components (Fig. S12d-f) are generally consistent with the trends in RSR (Fig. S12a-c), however some regional differences exist. For example, a strong increasing trend in the clear-sky atmospheric contribution rate is observed over equatorial western Pacific, which does not appear in its RSR. This is a moderating result of the decreasing contribution of the cloud component, indicating an increasing significance of the clear-sky atmospheric component for RSR in this region. In addition, the cloud component contribution rates show a wider distribution of increasing trends over the Arctic compared to the RSR. This is not only due to the increase in RSR from the cloud component, but also closely related to the significant decrease in the surface component at high latitudinal zones (Fig. 3i). This indicates that cloud components are playing an increasingly crucial role in the radiation budget in the Arctic (Sledd and L'ecuyer, 2021a; Sledd and L'ecuyer, 2021b).

There is no significant trend in the contribution of each latitudinal zone to the hemispheric RSR, except for a significant decreasing trend from 30°N to 40°N (Fig. 4d). Although the decreasing trend in cloud component of RSR at this latitude zone is greater than that of the clear-sky atmospheric component (Fig. 3i), the significant decreasing trend in the contribution rate of this latitude zone to the hemispheric RSR is mainly due to the reducing clear-sky atmospheric contribution. For the SH, trends in the different components cancel each other out, resulting in no trend in the contribution of the latitudinal zones to the total hemispheric RSR. For example, in the 0°-50°S region, the significant decreasing cloud component's contribution to the hemispheric RSR is offset by increasing clear-sky atmospheric and surface component

contributions in the hemispheric RSR. The opposite trend between the cloud component contribution and the clear-sky component contribution to some extent reflects the masking effect of clouds on clear-sky reflection. The reduction in clouds allows for a greater unmasking of the clear-sky component.

### 3.2 Can other radiation data exhibit hemispheric symmetry of RSR?

As mentioned in the introduction part, AVHRR, ISCCP, MERRA-2, and ERA5 can provide longer-term TOA RSR data compared to CERES EBAF. If these datasets can exhibit the hemispheric symmetry of RSR observed by CERES, it would greatly assist in identifying the underlying mechanism responsible for the hemispheric symmetry of RSR at longer time scales and exploring how the symmetry changes with time.

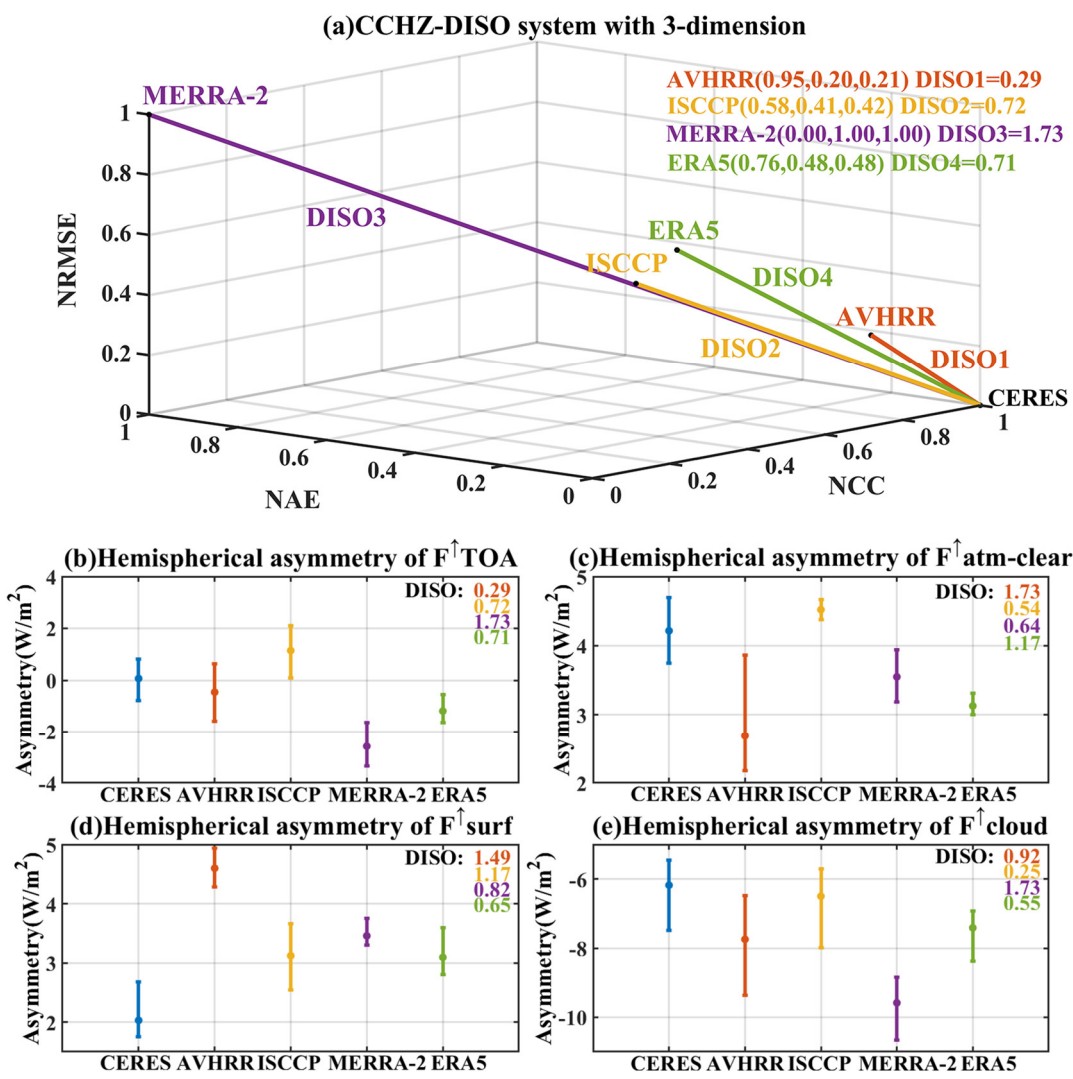

**Figure 5: (a) CCHZ-DISO system with 3-dimension for hemispheric difference of annual-average total RSR between NH and SH. The coordinate axis consists of three statistical indicators, normalized correlation coefficient (NCC) for x-axis, normalized absolute error (NAE) for y-axis, and normalized root mean square**

error (NRMSE) for z-axis. The DISO value is defined as the Euclidean distance between the three statistical indicators of each dataset and that of CERES (see Section 2.2.4 for details). Multi-year averages of hemispheric differences between NH and SH in (b) TOA RSR and its (c) clear-sky atmospheric, (d) surface, and (e) cloud components for the five datasets from Mar./2001-Feb./2016, as well as their DISO. with the maximum annual average difference for the dataset at the top of the error bars and the minimum at the bottom of the error bars. The blue, orange, yellow, purple and green bars indicate the statistical results for CERES EBAF, Cloud_cci AVHRR, ISCCP, MERRA-2, and ERA5, respectively. The numbers in the upper right corner are the DISO value of time series for hemispheric differences of different components for different datasets.

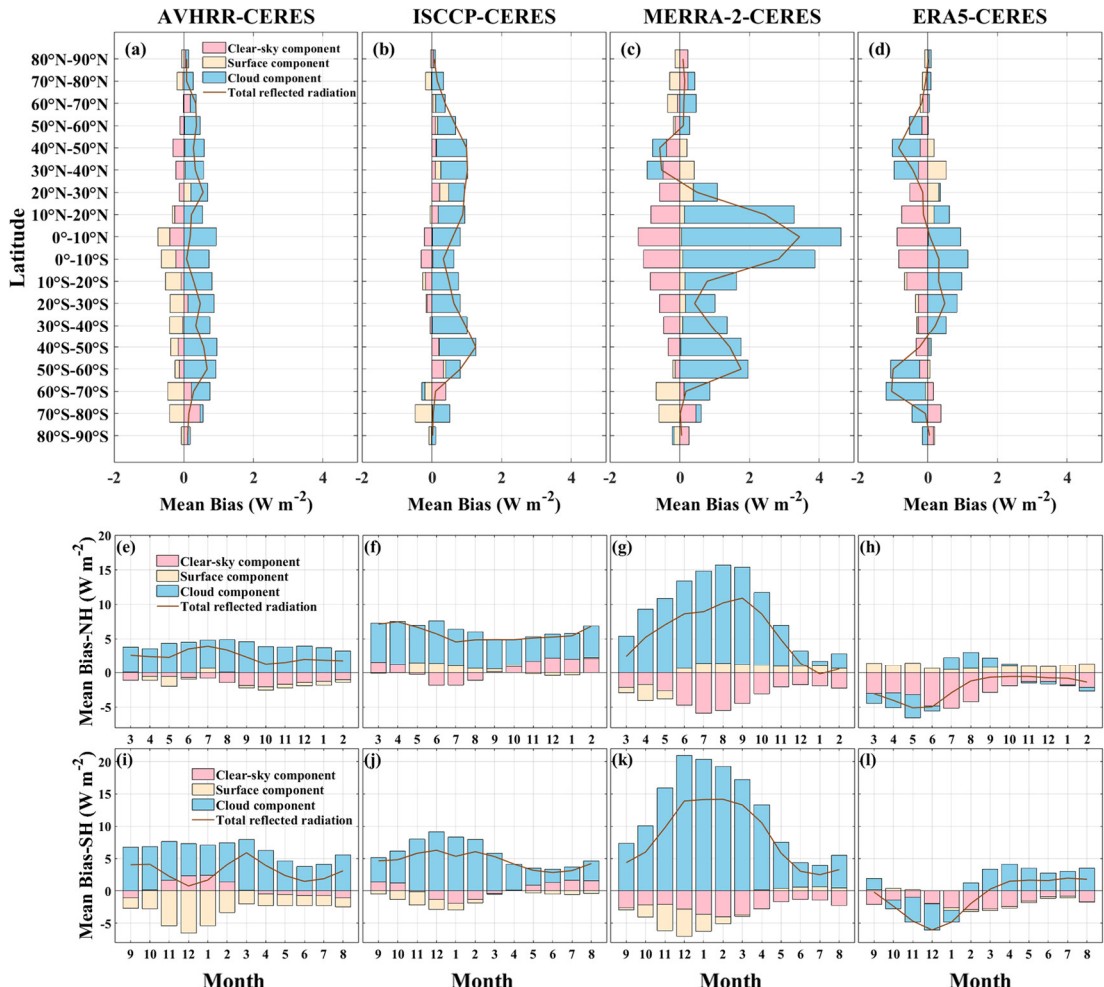

Figure 6: (a-d) Multi-year annual mean biases of total RSR and its components compared with CERES at different latitudinal zones for various datasets. Multi-year monthly mean biases of total RSR and its components compared with CERES in the (e-h) NH and (i-l) SH for different datasets from Mar./2001-Feb./2016. The columns from left to right represent the AVHRR, ISCCP, MERRA-2, and ERA5 datasets.

In order to comprehensively assess the performance of each dataset for hemispheric symmetry of RSR, Figure 5a presents three-dimensional results based on CERES EBAF data using the CCHZ-DISO data evaluation system. Figure5b-e further decompose the TOA total RSR of these datasets into clear-sky atmospheric, surface and cloud components, and compare the five datasets in terms of multi-year

averaged hemispheric asymmetry (NH-SH) of RSR and its components. Note that a dataset may perform well in hemispheric differences of RSR because a consistent positive or negative bias in both hemispheres, effectively offsetting between them. Additionally, the poor ability in reproducing hemispheric differences in RSR may also be attributed to biases in specific latitude zones and months. Therefore, we decompose the average biases of total RSR and its components compared with CERES for different datasets into latitude zones (Fig 6a-d) and monthly scales in NH (Fig 6e-h) and SH (Fig 6i-l) to further identity the potential error sources of in their reproduction performance of RSR hemispheric difference.

Different assessment metrics used for CCHZ-DISO system can produce different statistical results (Table S1). This means that we must select the most appropriate assessment metrics based on the specific research requirements to ensure the most applicable dataset. Note that the inclusion of spatial correlation coefficient in the DISO system did not notably alter the results (Table S1c), so the three recommended metrics (NCC, NAE, and NRMSE) are still used. In general, Fig. 5a indicates that AVHRR has the closest DISO value to CERES (DISO1=0.29) and exhibits the best performance in terms of hemispheric symmetry. It is followed by ERA5 (DISO4=0.71) and ISCCP (DISO2=0.72), while MERRA-2 performs the worst (DISO3=1.73). The DISO assessment metrics for the interannual series of hemispheric differences in total RSR and its component are shown specifically in Table S2.

Even in terms of multi-year average annual mean hemispheric differences (Fig. 5b), AVHRR is the closest to CERES. In fact, the remarkable ability of AVHRR to agree with the interannual hemispheric symmetry of RSR from CERES is attributed to its simultaneous slightly larger RSR in both hemispheres (Fig. 6a). Biases in hemispheric differences among different components cancel each other out, explaining this statistical result. It is clear that the hemispheric asymmetry of the three components of the AVHRR quite differs from that of CERES. Figure 5c shows that AVHRR has larger bias versus CERES in hemispheric asymmetry of the clear-sky atmospheric component, however, the bias of the clear-sky atmospheric component of AVHHR in different latitudinal zones is not as large as that of MERRA-2 and ERA5 (Fig. 6). The largest bias in hemispheric asymmetry and the highest DISO value for the clear-sky atmospheric component of AVHRR are mainly due to the fact that: (1) the clear-sky atmospheric component of AVHRR exhibits a certain bias versus CERES in NH but is minimal in SH, resulting in the interhemispheric bias not canceling each other out as observed in other datasets (Fig. 6a); (2) AVHRR also fails to capture the interannual variations in the hemispheric differences of the clear-sky

atmospheric component as observed by CERES and the data itself displays a high degree of annual dispersion. This ultimately leads to poor temporal correlation coefficients in the DISO calculations (CC=-0.52), resulting in its largest DISO value among all datasets for clear-sky atmospheric component (DISO=1.73). This bias of clear-sky atmospheric component between AVHRR and CERES are partly

due to the fact that the current version of AVHRR dataset assumes a fixed aerosol optical thickness (AOD) of 0.05. This assumption will underestimate the AOD under conditions of high aerosol loading, resulting in a bias in the radiative flux (Stengel et al., 2020). Furthermore, AVHRR exhibits the poorest performance in terms of hemispheric differences in surface components (Fig. 5d, DISO=1.49). Its multi-year average hemispheric differences of surface component is more than twice as large as that of CERES

(Fig. 5d), which mainly originate from the underestimation of the surface component by AVHRR only in the SH (Fig. 6a). It is mentioned in the ESA Cloud_cci Product Validation and Intercomparison Report (PVIR) that the Cloud_cci dataset exhibits higher biases in TOA RSR compared to CERES in regions with low vegetation coverage and typically high surface albedo. In terms of cloud component, AVHRR has slightly higher values than that of CERES in both hemispheres, particularly in the SH (Fig. 6a), thus

exhibits obvious bias from CERES in the hemispheric differences (Fig. 5e). Stengel et al. (2020) pointed out that AVHRR PMv3 shows a greater bias in identifying liquid clouds and reducing ice water paths compared to v2.

Although the overall performance of ISCCP in reproducing the hemispheric symmetry of total RSR is comparable to ERA5 (DISO_ISCCP=0.72; DISO_ERA5=0.71), it is the only dataset that exhibits a

695 brighter NH than SH, which is in agreement with CERES (Figure 5b), because its RSR in NH is more positively biased than its RSR in SH compared to CERES (Figure 6b). Additionally, its multi-year means of the hemispheric differences for all three components are closest to that of CERES among the datasets. However, its annual mean hemispheric difference of the surface component shows poor temporal-correlation with CERES (CC=0.25), thus exhibiting the larger DISO (DISO=1.17). On the other hand,

ISCCP performs best in reproducing the hemispheric differences of the clear-sky atmospheric component (DISO=0.54) and the cloud component (DISO=0.25). The inclusion of the Max Planck Institute Aerosol Climatology (MAC) in the treatment of stratospheric and tropospheric aerosols in the ISCCP-H series helps reduce the misidentification of aerosols as clouds (Young et al., 2018), thereby improving the simulation of clear-sky atmospheric components. Moreover, Fig. 6b shows that ISCCP has higher value

of the cloud component in both hemispheres than that of CERES. The offsetting effect results in a hemispheric difference in cloud component that is closest to CERES.

The two reanalysis datasets exhibit quite different performance in simulating hemispheric differences in the total RSR and its three components. Among all the datasets, MERRA-2 has the largest DISO value relative to the hemispheric difference in RSR observed by CERES, implying that it performing worst (Fig. 5a). This may be primarily influenced by cloud cover bias (Lim et al., 2021). Indeed, MERRA-2 poorly represents the hemispheric difference in the cloud component (DISO=1.73), whereas ERA5 shows better agreement with CERES (DISO=0.55). The latitudinal distribution of the RSR bias reveals that although the negative bias of the clear-sky atmospheric component partly offsets the great positive bias of the cloud component, the total RSR bias of MERRA-2 is still the largest in all datasets, especially in SH (Fig. 6c). Hinkelman (2019) pointed out that the difference of all-sky RSR at TOA between MERRA-2 and EBAF is attributed to differences in cloud variables such as cloud fraction or optical depth. This bias may stem from a flaw in the cloud parameterization (e.g., cumulus parameterization and convective cloud schemes) within the reanalysis assimilation model (Dolinar et al., 2016; Li et al., 2017). Besides, MERRA-2 also exhibits large biases in the clear-sky atmospheric component in different latitudinal zones compared to other datasets (Fig. 6c), but the bias of its hemispheric asymmetry is smaller due to the inter-hemispheric cancellation. Nevertheless, ERA5 still exhibits good consistency with CERES, except for the hemispheric difference in the clear-sky atmospheric component (DISO=1.17). Li et al. (2023) demonstrated that the deviation of ERA5's surface solar radiation products from observed values increases with higher aerosol loading, indicating that aerosols highly affect the accuracy of ERA5's radiation products, which may affect the calculation of the clear sky component.

Compared to other datasets, AVHRR exhibits smaller positive biases versus CERES in the multi-year latitude-zone-averaged total RSR (Fig. 6a). It is a result of the widespread positive bias of cloud components across all latitude zones globally, which is offset by the negative bias of clear-sky atmospheric components in the NH and surface components in the SH. In the NH, the AVHRR data may misidentify high aerosol loads as clouds, thus exhibiting smaller clear sky atmospheric components in NH with rich dust and anthropogenic aerosol and larger cloud component (more details are described in PVIR). The surface component in the SH shows a large negative bias compared to that of NH. This is

mainly due to the different surface albedo retrieve algorithms and input surface parameters of AVHRR

for land and ocean (more details are described in "ESA Cloud cci Algorithm Theoretical Baseline

Document v6.2"). Compared to CERES, AVHRR has larger surface albedo of land at low and middle

latitudes and smaller surface albedo of the oceans and polar regions (Fig. S13). This is why the surface

component of AVHRR exhibits a great negative bias in the SH compared to CERES. In the NH, the

surface component biases for land and ocean cancel each other out and therefore contribute little to the

total RSR bias. From a monthly scale perspective, the positive biases of cloud components by AVHRR

are present in all months in both hemispheres. The negative biases of clear-sky atmospheric components

in the NH are particularly pronounced during autumn and winter (Fig. 6e), while the negative biases of

surface components in the SH are largest in November, December, and January (Fig. 6i), which are

related to the seasonal variation in the incident solar variation and surface albedo biases.

Compared to CERES, ISCCP exhibits the largest mean bias in the 40°-50° latitude zones in both

hemispheres, primarily driven by the positive bias of cloud component (Fig. 6b). The ISCCP data

combines observational data from geostationary satellites in low- and mid-latitude regions, thus the

higher viewing zenith angle compared to low-latitude regions introduces greater uncertainty in the

retrieval of cloud fractions in mid-latitude regions (Evan et al., 2007; Marchand et al., 2010; Norris and

Evan, 2015; Boudala and Milbrandt, 2021), consequently resulting in larger cloud component biases.

Boudala and Milbrandt (2021) found that ISCCP overestimates cloud cover between approximately 40°

and 60° latitudes in both hemispheres, particularly in North America and Europe. In the NH, the bias

from cloud component shows no clear seasonal variation (Fig. 6f) as in the SH, which is larger from late

spring to summer (Fig. 6j). Although ISCCP demonstrates minimal average bias in surface components

across almost latitude zones except for 70°S-80°S (Fig. 6b), its combined performance in hemispheric

differences of surface component is relatively poor (DISO=1.17). This is because DISO is a

comprehensive assessment based on three metrics (NCC, NAE, NRMSE), whereas ISCCP's hemispheric

difference of surface components exhibit poorer temporal correlation with CERES (CC=0.25), indicating

its limited ability to capture the interannual variations of surface components.

For the MERRA-2, the zonal-averaged total RSR exhibit most pronounced biases compared to other

datasets, particularly positive mean bias in the latitude 0-20° in both hemispheres and 30°S-60°S,

primarily attributed to a considerable positive bias of cloud component (Fig. 6c). Previous study also

pointed to excessive cloud cover over tropical oceans and the Southern Ocean has in MERRA-2 (Hinkelman, 2019). The lack of cloud and radiation-related data assimilation also have introduced uncertainties in the simulated RSR in MERRA-2 (Yao et al., 2020). The large positive bias in cloud component in the mid-latitudes of the SH may be due to the fact that MERRA-2 overestimates the frequency of supercooled liquid clouds over the Southern Ocean during the summer (Kuma et al., 2020). Furthermore, its negative biases of clear-sky atmospheric components are mainly concentrated in the low- and mid-latitudes, especially in the tropics, and exhibit interhemispheric symmetry. This partially explains its better performance in reproducing hemispheric differences of clear-sky atmospheric components, as the biases between hemispheres can offset each other. The inability to effectively distinguish cloudy and clear-sky conditions for high aerosol loadings scenarios (Trolliet et al., 2018) and the lack of emission data in the aerosol model of MERRA-2 (Buchard et al., 2017) may lead to a large underestimation of high AOD values, hence lower clear-sky atmospheric components. Additionally, the RSR bias from MERRA-2 also shows notable monthly variations (Fig. 6g and 6k). On the one hand, it links to the seasonal variation of incident solar radiation mean biases, while the temporal correlation of between RSR mean biases and incident solar radiation is 0.68 and 0.64 in NH and SH, respectively. On the other hand, the positive mean biases of RSR are driven by the positive cloud component biases. The positive cloud component bias in the NH reaches over 10 W m$^{-2}$ from May to October, with the peak in late summer (August, bias=14.29 W m$^{-2}$), while in the SH, the bias generally exceeds that of the NH from October to April, with the peak in early summer (December, bias=20.94 W m$^{-2}$). Moreover, MERRA-2 has a large negative bias of surface components compared to CERES in Antarctica during melting season (November to January), which could be due to biases in the input snow products that introduce remarkable uncertainties in surface albedo (Jia et al., 2022).

For the ERA5, the total RSR between 10°N and 40°S is higher compared to that of CERES, while at other latitudes the RSR is lower, which primarily driven by cloud component biases (Fig. 6d). Previous research indicated that ERA5 systematically overestimates high cloud fraction in the tropical convective regions (Wright et al., 2020) while underestimating liquid and ice water paths of clouds in the Arctic (Jenkins et al., 2024). In terms of the hemispheric monthly biases, the positive bias of cloud component in the SH mainly occurs during the autumn and winter seasons (Fig. 6l). Apart from the high latitudes in the SH, ERA5 shows negative biases of clear-sky atmospheric components across all latitude zones,

especially in the tropics, which may be attributed to inadequate representation or simulation of aerosols and aerosol-cloud interactions in ERA5 (Jiang et al., 2020). This may be related to the shortcomings of ERA5's aerosol assimilation process, which only considers aerosol climatology as input, overlooking

aerosol variations on interannual time scales (He et al.,2021). Surprisingly, apart from positive biases in the 20°N-50°N region, the multi-year averaged surface component of ERA5 basically the same as CERES. Jia et al. (2022) also pointed that ERA5 captures changes in snow albedo at mid and high latitudes better than other reanalysis data.

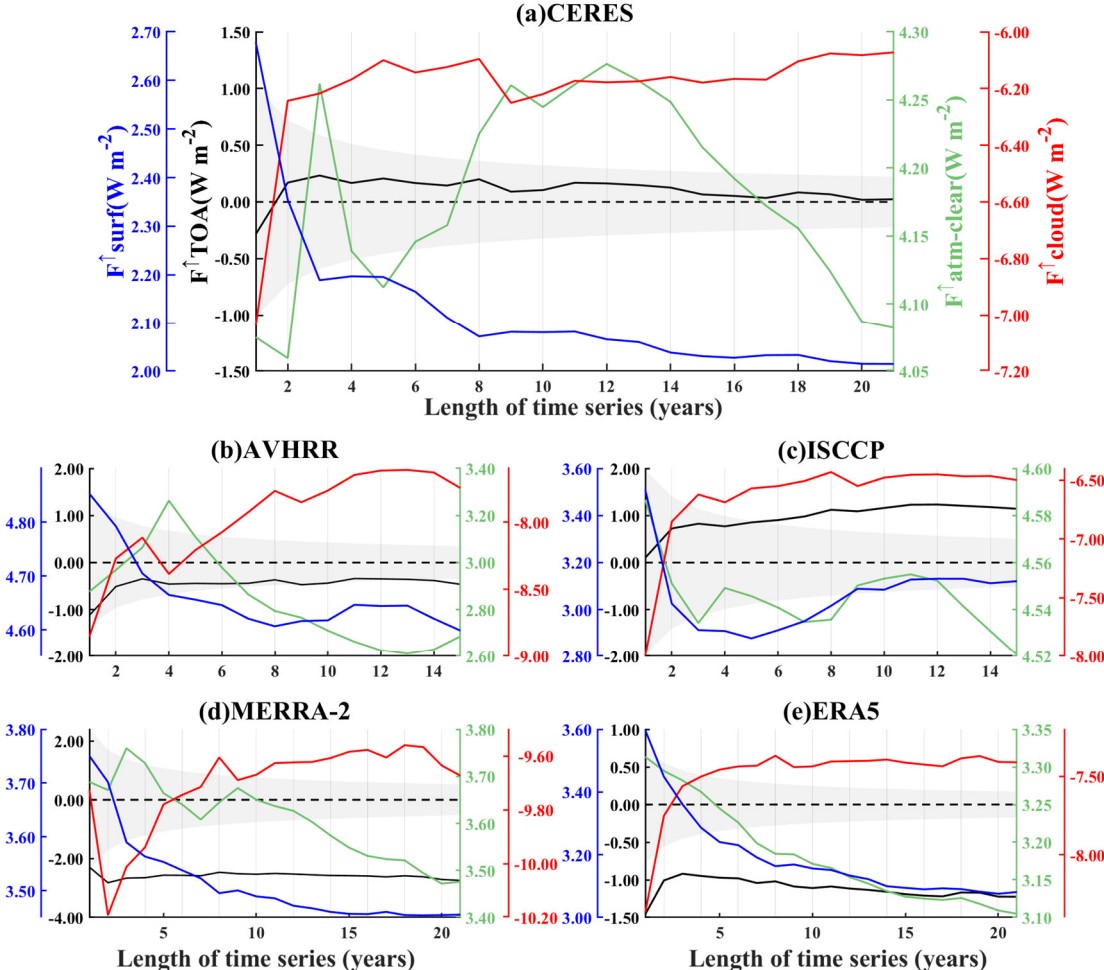

**Figure 7: Cumulative annual mean for hemispheric differences of RSR and its components for (a) CERES, (b) AVHRR, (c) ISCCP, (d) MERRA-2 and (e) ERA5. That is, when Length of time series (years) = N, the hemispheric differences (NH-SH) of annual mean RSR are calculated from 2001 to 2000+N.The range of N varies due to the different record lengths of the datasets, with 1≤N≤21 for CERES,MERRA-2 and ERA5, while for AVHRR and ISCCP, 1≤N≤15.The black colour indicates the hemispheric difference of the total RSR,**

**while the blue, green, and red colours correspond to the hemispheric differences of the three components, respectively (as y-axis labels in a).The shaded areas are the uncertainties of hemispheric difference of RSR for the given dataset. If the solid black line is within the shaded area, it indicates that the hemispheric symmetry in total RSR is credible within the uncertainty.**

In a recent study, Datseris and Stevens (2021) pointed out that the symmetry of albedo cannot be established on an annual or sub-annual scale, but rather on larger spatial and temporal scales. It prompts us to find out what time scale for these datasets can be used for the study of hemispheric symmetry of PA. In following analysis, we use radiation datasets from different sources to investigate the appropriate time scale for studying the hemispherical symmetry of RSR. Figure 7 illustrates the variation of multi-year average hemispheric differences of RSR and its components over the cumulative length of the time series (N) for different datasets, i.e. the N-year averaged hemispheric difference of RSR. Figure 7a shows that the hemispheric differences in total RSR and its components observed by CERES are tending to stabilize over time, except for the clear-sky atmospheric component. The hemispheric asymmetry of clear-sky atmospheric component exhibits a strong perturbation over time, which may be closely related to human activities or natural perturbations, particularly the highly variable emissions of anthropogenic aerosols and irregular occurrences of large-scale volcanic eruptions and forest fires (Minnis et al., 1993; Diamond et al., 2022). Although previous research, including our own study, have demonstrated that CERES observes nearly equal RSR at the TOA in both hemispheres, there is no fixed quantifiable measure to define this "nearly equal" condition. Therefore, we try to discuss various numerical or criteria for assessing the symmetry of RSR between hemispheres here. Voigt et al. (2013) conducted a random division of the Earth into two halves to assess whether these random pairs exhibited hemispheric symmetry in RSR. The results revealed that only 3% of the random pairs demonstrated a hemispheric difference in RSR smaller than 0.1 W m$^{-2}$, as measured by CERES-EBAF. Furthermore, even when this criterion was extended tenfold (1 W m$^{-2}$), only 31% of the random pairs satisfied the hemispheric symmetry requirement. Stephens et al. (2015) noted that the multi-year averaged hemispheric difference in RSR between the NH and SH is less than 0.2 W m$^{-2}$, suggesting this as an indicator of hemispheric symmetry. Here, when we use a symmetry criterion of 0.1 W m$^{-2}$, CERES achieves hemispheric symmetry of RSR on a 15-year annual mean scale, while none of the other datasets do. When we expand this symmetry criterion to 0.2 W m$^{-2}$, the symmetry study application of CERES is around 9-year scale, and other datasets remain inapplicable. When held to a more conservative standard of 1 W m$^{-2}$, CERES achieves hemispheric symmetry every year, and AVHRR achieves it on scale of two years. Interestingly, the ISCCP exhibits increasing hemispheric asymmetry as the time span extends, only declining after a 13-year average. Similar, ERA5 also displays a similar but more moderate increase in hemispheric

asymmetry.

In addition, in order to have a more rigorous standard, the study takes the uncertainty of the instrumental measurements into account. That is, if the RSR difference between NH and SH is within the uncertainty of the measurement, it is considered as hemispheric symmetry (Diamond et al., 2022). The regional averaged (1° x 1°) monthly mean uncertainty of the RSR at the TOA from the CERES EBAF is 2.5 W m$^{-2}$ (Loeb et al., 2018b). Considering CERES as the true values, the monthly regional mean biases of AVHRR, ISCCP, MERRA-2 and ERA5 are 3.3 W m$^{-2}$, 4.8 W m$^{-2}$, 5.9 W m$^{-2}$ and -1.9 W m$^{-2}$, respectively, which will be used to calculate their uncertainties. Here we follow the method of Jönsson and Bender (2022) to calculate the uncertainty of hemispheric difference of RSR. Here, it is noting that only rough calculations have been made due to the unavailability of uncertainties at different grid points around the globe. Uncertainty in the time-mean over the N-month period is scaled by a factor of $N^{-1/2}$. Then there is a time series of the uncertainty in the hemispherical differences of RSR for each dataset. It is clear that as time grows, the range of uncertainty shrinks. Note that if the solid black line falls within the shaded area (see Figure 7), it indicates that the RSR exhibits credible hemispheric symmetry within the given uncertainty. It is clear that the hemispheric difference of the total RSR from CERES remains well within its uncertainty range. Similarly, AVHRR stays well within its uncertainty over a 14-year timescale. But, ISCCP only keeps within uncertainty on timescales up to 5 years. The reanalyzed datasets clearly deviate from their respective uncertainty ranges.

In summary, AVHRR shows better agreement with CERES in terms of the hemispheric symmetry of RSR. Furthermore, the cumulative annual mean time series of hemispheric differences in the cloud component for ISCCP and ERA5 display similar variations to CERES (CC_ISCCP=0.96; CC_ERA5=0.95), while ISCCP exhibits a smaller bias (AE_ISCCP=0.41; AE_ERA5=1.29). However, in the term of the cumulative annual mean time series of hemispheric differences for the surface component, only ISCCP doesn't agree with the observed variation in CERES, and its CC with CERES is 0.5 (insignificant), while the CCs of the other datasets are all greater than 0.95. For the cumulative annual mean of hemispheric differences in the clear-sky atmospheric component, although AVHRR, ISCCP, and MERRA-2 show similar abrupt variability patterns to CERES, indicating the irregularity of human and natural activities, they do not correlate well with CERES, with CCs of -0.63, -0.38, and 0.25, respectively. In contrast, ERA5 shows a continuous decrease trend, and correlates poorly with CERES,

with a CC of -0.24, which also verifies its poor modelling ability in the clear-sky atmospheric component.

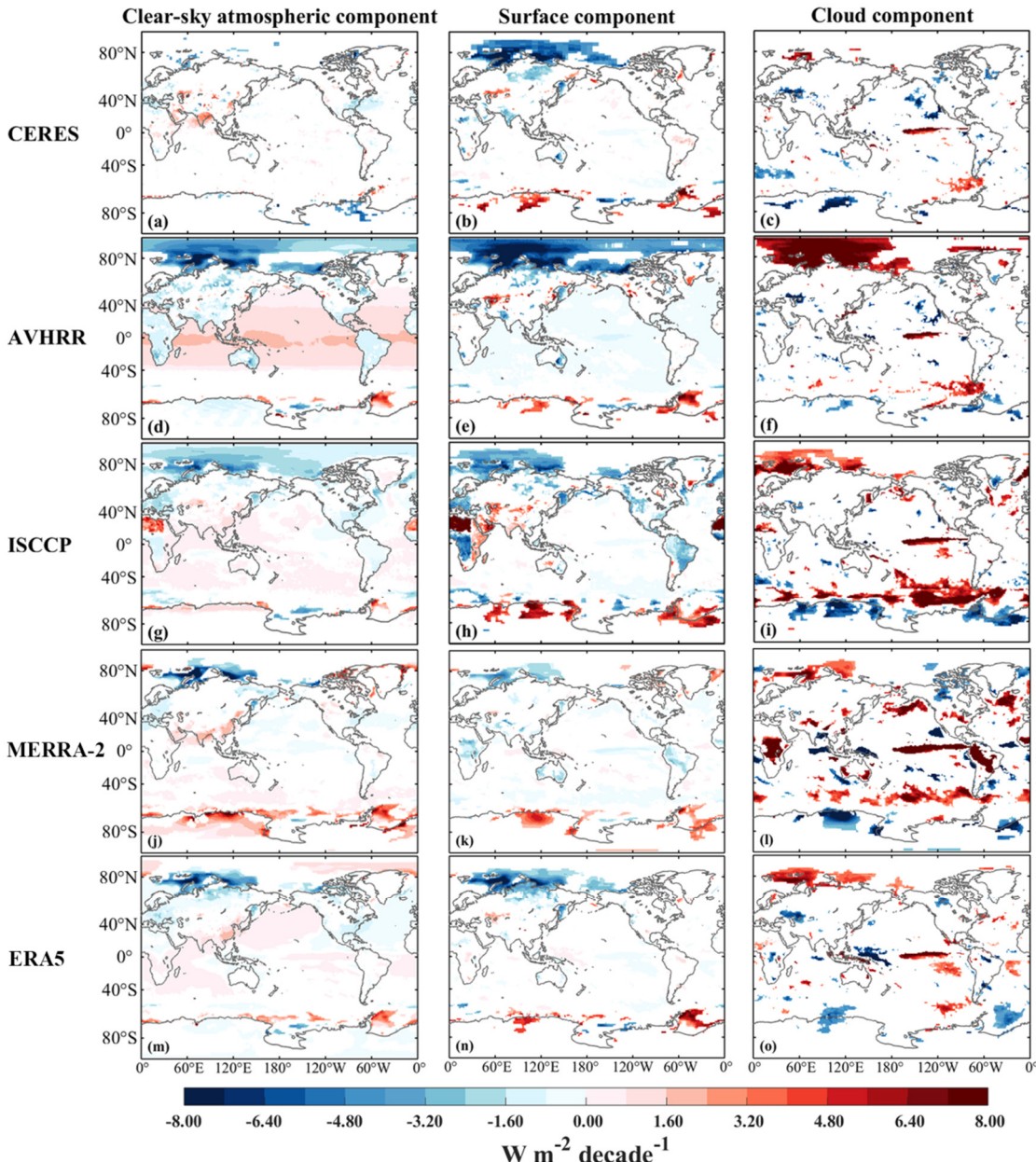

**Figure 8: Trends in TOA RSR flux of the clear-sky atmospheric component (left column), surface component (centre column) and cloud component (right column) for Mar./2001-Feb./2016. (a-c) CERES, (d-f) AVHRR, (g-i) ISCCP, (j-l) MERRA-2, and (m-o) ERA5.**

Figure 8 further illustrates the global distribution of the long-term trends in the RSR of three components from CERES, AVHRR, ISCCP, MERRA-2 and ERA5. Note that the trend analysis is based on de-seasonalized monthly time series from March 2001 to February 2016.

For the regional trends of clear-sky atmospheric components, there are considerable differences among the five datasets. Compared to CERES, the other four datasets exhibit some spurious trends over

the oceans, especially AVHRR. The AVHRR clear-sky atmospheric component shows a clear positive trend over the oceans within 40° of latitude in both hemispheres, which is not seen in other datasets. Analysis of the 1995-2015 time series of the clear-sky atmospheric component (Fig. S14 a) for the grid over the equatorial Pacific Ocean (179.5°E, 0.5°S) shows a decreasing trend in the clear-sky atmospheric component during the period 1995-2000, with a jump in March 2001, coinciding with the transition from NOAA14 to NOAA16. The jump in data triggers spurious trends, which are also evident in the upwelling shortwave radiative fluxes at the TOA and surface, respectively (Fig. S14 b, c). The changes in the March 2001 surface and TOA upwelling shortwave radiation fluxes may be attributed to an increase in the solar zenith angle (SZA) due to satellite orbital drift (Ji and Brown, 2017). The SZA is a key parameter in the surface bidirectional reflectance distribution function (BRDF) used by AVHRR_PM_V3, which affects the surface reflected energy. Increased SZA may trigger the Fresnel effect and an increase in the atmospheric path length, respectively, hence affecting the reflected energy. Note that orbital drift-induced changes in SZA are most pronounced at low latitudes (Privette et al. 1995), which may explain the latitudinal dependence of these spurious trends. Furthermore, the difference in the trend distribution under clear-sky conditions between AVHRR and CERES may be partly due to different methods of estimating the mean clear-sky fluxes. The clear-sky radiative fluxes of CERES are based on clear-sky conditions only (and interpolate the collected clear-sky radiative fluxes to cloudy pixels), whereas AVHRR takes into account all the conditions (but removes the clouds) (Stengel et al., 2020). ISCCP, MERRA-2, and ERA5 all capture a significant positive trend over India well, whereas AVHRR shows the opposite trend. Over the Arctic, both AVHRR and ISCCP show widespread of significant negative trends, which are not obviously seen in CERES and reanalysis data. ISCCP suggests a significant positive trend in clear-sky atmospheric component over North Africa, which is not presented by other datasets. For trends of surface component, the ERA5 is relatively consistent with CERES over the land but exhibits more spurious signals over the oceans. Despite some similarities in trend distribution of surface components on land between AVHRR and ERA5, there are widespread spurious decreasing trends similar to that of clear-sky atmospheric components over Arctic for AVHRR. All datasets show significant negative trends over the Arctic, but with different magnitudes and ranges. ISCCP shows significant positive trends of surface component over Central Africa, with opposite trends in North Africa and South Africa. These anomalous trends may be influenced by geometry artifacts observed by satellites. The

ISCCP dataset uses input parameters from a series of geostationary satellites, and the edges of satellite views may generate spurious variability (Evan et al., 2007). We selected a gird in North Africa with the strongest positive trend and examined its de-seasonalized monthly anomaly time series (Fig. S15). There is a sudden increase of RSR in July 2006 and since then there has been a persistent positive anomaly. This abrupt change explains strong trend in the RSR component in the African region. We speculate that this may be attributed to a sudden change in the geostationary observation platform (Evan et al., 2007). Over South America, ISCCP and MERRA-2 exhibit significant negative trends, which are not observed in other datasets. In addition, snow cover is a major source of error in surface albedo in reanalysis data (Jia et al., 2023). This could be a key reason for MERRA-2's failure to capture the declining trend in surface components in northern Russia. For trends of cloud component, all datasets find a significant increase over the equatorial central-eastern Pacific. However, except for AVHRR, the other datasets fail to capture the negative trend near the east Pacific adjacent to North America. Furthermore, compared to CERES, AVHRR and ISCCP have produced many trends not seen in CERES over polar regions. And every dataset mis-estimates the trend values in most regions. This indicates that the cloud retrieval algorithms for satellite-based datasets, as well as the cloud parameterization schemes for reanalysis datasets, which are key sources of uncertainty in their cloud components, still require improvement.

In terms of interannual hemispheric trends of RSR and its components (Table S3), all four datasets fail to capture the decreasing trend in total RSR for both hemispheres, and ISCCP and MERRA-2 even show an increasing trend in both hemispheres. For three components, AVHRR exhibits a greater positive trend in clear-sky atmospheric component and a negative trend in surface component in the NH. On the contrary, ISCCP, MERRA-2, and ERA5 fail to represent the decreasing trends in cloud components for both hemispheres and even show opposite trends.

In summary, if the focus of study is solely on the long-term changes in hemispheric symmetry of total RSR at TOA, AVHRR is the preferred choice. However, it is not recommended to use AVHRR for decomposing the RSR into components. Additionally, ISCCP can be used to investigate long-term hemispheric asymmetry changes and its mechanisms in cloud component.

## 4    Discussion and Summary

The hemispheric symmetry of RSR is a robust feature of the Earth-atmosphere system, and the

mechanisms by which it is currently maintained remain unclear, posing a great challenge for improving

the simulation of hemispheric symmetry of RSR in climate models. Several possible compensatory

mechanisms have been proposed, which are not only limited by latitude but also have seasonal

characteristics. If we resolve the energy down to monthly scales and latitudinal zones, we can gain insight

into the changes of RSR at finer spatial and temporal scales and further improve the understanding of

potential regional-scale mechanism for hemispheric symmetry of RSR. In addition, we also evaluate the

applicability of radiation datasets with longer records in studying hemispheric symmetry over time. The

main findings are as follows:

(1) RSR shows a decreasing trend in both hemispheres across almost all months and all latitudinal

zones, with differing primary driven factors. In the NH, the interannual hemispheric decreasing trend is

jointly influenced by decreasing trends of the three components, while in the SH, only the cloud

component exhibits a significant decreasing trend. Monthly trends indicate a slowdown in the decreasing

trend from spring to winter, with the maximum trend occurring in the spring (April in NH and October

in SH). For the NH, most of the downward trend in RSR originates from 30°N-50°N, with extremes in

the 30°N-40°N. At 30°N-50°N, the trend is attributed to a significant decrease in both the cloud and clear-

sky atmospheric components. The decreasing in the clear-sky atmospheric component is due to reduced

emissions of anthropogenic sulfate aerosols from various regions and a weakening of dust activities

during spring and summer in parts of the dust belt. The decreasing trend in the cloud component is

concentrated near the eastern Pacific and North Atlantic close to North America, which may be related

to a decrease in low cloud cover and optical thickness, respectively. Specifically, the decrease in low

cloud cover in the eastern Pacific is attributed to the increase in SST, which may be related to the shift

of the PDO phase from negative to positive. For the SH, the significant decreasing trends of RSR is

mainly occur in the 0°-50°S, which is entirely dominated by the significant decreasing trend in the cloud

component. This reduction in cloud component is mainly observed over the south tropical western Pacific

as well as over the wider Southern Ocean, attributed to the reduction in cloud cover. Unlike the three

components of RSR in the NH, there is no significant trend in the proportion of their contribution rates,

indicating that there is no significant adjustment in the radiation budget in the NH. The contribution rate

of the clear-sky atmospheric component in the SH is increasing, while that of the cloud component is

decreasing. Notably, the rate of contribution of the total RSR in 30°N-40°N latitude zone to the

hemispheric RSR has significantly decreased, primarily due to a reduction in the contribution of the clear-sky atmosphere component.

(2) According to the CCHZ-DISO assessment system, AVHRR performs best hemispheric symmetry of RSR, followed by ERA5 and ISCCP, and the worst is MERRA-2. The better performance of AVHRR in hemispheric difference of RSR is due to its simultaneous slight positive biases of both hemispheres, driven by offsetting biases in different components. While AVHRR performs worst in capturing the hemispheric difference of clear-sky atmospheric and surface components, its component

biases in different latitude zones are in fact smaller than those of other datasets, except that they are asymmetric and therefore do not offset between two hemispheres. In contrast, ISCCP performs best in reproducing CERES-observed hemispheric differences of clear-sky atmospheric and cloud component, but shows positive bias in the cloud component in the mid-latitudes, possibly influenced by the field of view of geostationary satellites. The total RSR bias between MERRA-2 and CERES is mainly

concentrated in the 20°N-20°S and 40°S-60°S, with extreme values in the summer, dominated by the large overestimation of cloud components. ERA5 is the best dataset for reproducing hemispheric difference of surface component, and is in excellent agreement with CERES in the SH. Under different symmetry criteria, the applicability of different datasets to hemispheric symmetry of RSR studies vary. CERES can achieve hemispheric symmetry at a 15-year average with the 0.1 W m$^{-2}$ criterion, and when

the criterion is extended to 0.2 W m$^{-2}$ and 1 W m$^{-2}$, the years of applicability are advanced to 9-year and every year. AVHRR can achieve hemispheric symmetry within its uncertainty of the 14-year time scale. ISCCP achieves hemispheric symmetry within its uncertainty on a 5-year scale, but shows increasing hemispheric asymmetry over time. Both reanalysis datasets are far from the criterion of hemispheric symmetry of RSR. All datasets fail to capture the changes in multi-year averaged hemispheric differences

of clear-sky atmospheric components as the record length increases, possibly due to a lack of data assimilation for anthropogenic aerosol emissions and large-scale biomass burning activities. In addition, all datasets struggle in capturing hemispheric and regional trends in RSR and its components.

    Based on long-term satellite observations, this study and previous research have confirmed a clear decreasing trend in solar radiation reflected back into space in both hemispheres over the past two

decades (Loeb et al., 2020; Stephens et al., 2022). In addition, previous study (Loeb et al., 2022) pointed out that a significant increasing trend of LW radiation emitted to space is found in the NH, while no

significant trend is observed in the SH. Loeb et al. (2021b) noted that the increase in outgoing LW radiation is primarily due to the increasing global surface temperature and changes in clouds, although it is partly compensated by the increase in water vapor and trace gases. However, the overall increase in

outgoing LW radiation does not offset the decrease in RSR, resulting in a positive trend in the net radiative flux in both hemispheres (indicating that the Earth is absorbing more energy) (Raghuraman et al., 2021). This positive trend in the Earth's energy imbalance (EEI) will exacerbate global warming, sea-level rising, increased internal heating of the oceans, and melting of snow and sea ice (IPCC, 2013; Von Schuckmann et al., 2016; Loeb et al., 2021b; Forster et al., 2021). Indeed, a recent study based on long-term

homogenized radiosonde data indicated that the atmosphere has become more unstable in the NH during the period 1979-2020 (Chen and Dai, 2023). Given the profound impact of these changes on the climate system, it is crucial to pay closer attention to the future evolution of PA and its symmetry. Although climate models persistently exhibit biases in simulating the mean state of albedo symmetry from CMIP3 to CMIP6 (Crueger et al., 2023), they remain a powerful tool for generating hypotheses about the

unexplained observed RSR symmetry (Rugenstein and Hakuba, 2023) and projecting future evolutions and potential influencing mechanisms. For example, Rugenstein and Hakuba (2023) examined the response of modeled surface temperature and RSR to $CO_2$ and found an increasing difference in surface warming between the two hemispheres under stronger carbon dioxide forcing and weaker aerosol forcing. They also proposed that the warmer hemisphere will become darker, suggesting a potential asymmetry

in albedo in the coming decades. On the other hand, Diamond et al. (2022) focused on changes in clear-sky hemispheric asymmetry under different emission scenarios simulated by their model. Their results indicated a significant shift in clear-sky albedo asymmetry throughout this century under both high and low emission scenarios, primarily driven by anthropogenic aerosol emissions and cryosphere changes. Furthermore, Jönsson and Bender (2023) investigated the evolution of hemispheric albedo differences

following a sudden quadrupling of $CO_2$ concentration using CMIP6 coupled model simulations. They found that the initial albedo reduction in the NH may be partly compensated by a reduction in extratropical cloudiness in the SH on a much longer timescale which can be referred to as a mechanism of trans-hemispheric communication. They also highlighted that if RSR maintains hemispheric symmetry, compensating for cloud variations will have uncertain but important effects on Earth's energy balance

and hydrologic cycle. However, whether the hemispheric symmetry of RSR can be sustained indefinitely

remains an open question. Therefore, it is essential to focus on investigating additional potential mechanisms of hemispheric RSR symmetry and future projections using model ensembles, along with observational constraints.

**Data availability.** The CERES_EBAF_Ed4.2 product is publicly available through the NASA Langley Research Center CERES ordering tool at https://ceres.larc.nasa.gov/data/. The ESA Cloud-cci version 3 products, AVHRR-PMv3 for this research are included in the paper: Stengel et al. (2020), or obtained through https://public.satproj.klima.dwd.de/data/ESA_Cloud_CCI/CLD_PRODUCTS/v3.0/L3C/. The ISCCP-FH data are available from the following website: https://isccp.giss.nasa.gov/pub/flux-fh/. The

MERRA-2 dataset used in this study is available from the following websites: https://doi.org/10.5067/OU3HJDS973O0. The ERA5 monthly averaged data on single levels from 1940 to present are available from Climate Data Store (CDS) of https://cds.climate.copernicus.eu/cdsapp#!/dataset/reanalysis-era5-single-levels-monthly-means?tab=overview.

**Author contributions.** RL and BJ organized the paper and performed related analysis. RL drew article graph and prepared the manuscript. BJ and JL conceptualized the paper and revised the whole manuscript. DW, LZ, YW and YW modified the paper and provided suggestions for this study. All authors contributed to the discussion of the results and reviewed the manuscript.

**Competing interests.** The contact author has declared that none of the authors has any competing interests.

**Disclaimer.** Publisher's note: Copernicus Publications remains neutral with regard to jurisdictional 1045 claims in published maps and institutional affiliations.

**Acknowledgements.** We would like to thank the CERES, CLOUD_CCI AVHRR, ISCCP, MERRA-2, and ERA5 science teams for providing excellent and accessible data products that made this study possible. We sincerely thank the editor and referees (Aiden Jönsson and two anonymous referees) for

taking the time and effort necessary to review the manuscript. We sincerely appreciate all valuable comments and suggestions, which helped us to improve the quality of the manuscript. We would like to acknowledge the assistance of ChatGPT in proofreading and polishing the language of this manuscript. ChatGPT, an AI language model developed by OpenAI, helped refine the clarity and fluency of the text. We are grateful for its contribution in improving the quality of our work.

**Financial support.** This research was jointly supported by the National Science Fund for Excellent Young Scholars (42022037), the Major Program of the National Natural Science Foundation of China (42090030), the National Natural Science Foundation of China (42305072), and the China Postdoctoral Science Foundation (2023M731454).

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
