# Peer review of "Understanding the variation of Reflected Solar Radiation: A Latitude- and month-based Perspective"

_EGUsphere, 2023_

## Author Comment (AC1)

**Response to Reviewer #1's Comments:**

Ruixue Li et al. (Author)

**We are very grateful for the Reviewer #1' detailed comments and suggestions, which help us improve this paper significantly. Based on the comments and suggestions from reviewer, we reorganize the abstract, introduction, datasets and methods, results and conclusion sections to highlight the innovations, key points of this study and improve coherence between sections. In addition, we add some interpretations and delete some superfluous information in each section in order to make the manuscript clearer.**

**Important revision includes:**

(1)  Section 1, Introduction, is rewritten to enhance clarity regarding the significance and innovations of the manuscript. Superfluous information is removed, and specifically, we highlight the reasons for studying the changes in reflected solar radiation (RSR) at finer spatiotemporal scales and the importance of assessing the applicability of different radiation datasets with longer record in hemispheric symmetry studies. More details are described in Line 65-138. The abstract is also further reorganized to summarize the main findings of this paper.

(2)  In Section 2, Datasets and Methodology, we update the CERES data to the latest version (CERES-EBAF 4.2) and correct some bugs in the data processing, and revise the corresponding descriptions.

(3)  We agree the reviewer's comments. In Section 3.1, we acknowledge that there is a little bit overlap between Fig. 1 and previous studies. However, the focus of this section is on analyzing changes in RSR and its components at the latitude and monthly scales. We have reorganized Section 3.1 and added an expanded discussion on hemispheric RSR trends, highlighting the contributions of finer spatiotemporal scales to these trends. Additionally, we have analyzed the changes in the contribution rates of different components to the RSR.

(4)  The content discussing the contribution of different factors to latitudinal zones in extreme years (Section 3.2) is removed. The adjustment allows us to maintain a more focused and streamlined discussion, centering on the primary objectives and findings of the study.

(5)  We substantially expand the original Section 3.3 (now revised as Section 3.2) to systematically quantify the performance of four radiation datasets in reproducing hemispheric differences and symmetry

of RSR observed by CERES at hemispheric and finer temporal-spatial scales. This expansion ensures a smoother and tighter connection with Section 3.1.

(6) In the Section 4, Discussion and Summary, we revise the main conclusions of this study and delete some superfluous information.

(7) In each section, we also add some interpretations about the comments from reviewers.

**Please see our point-by-point reply to comments. All revisions are shown in revised manuscript by using track changes.**

**General responses:**

1. The authors cite large amounts of published studies on hemispherical albedo symmetry. The authors analyze southern and northern hemispheres' surface and atmosphere contributions. However, the result of the zonal analysis is mainly that higher land albedo in the northern hemisphere present in the mid-latitude is compensated by clouds over southern ocean, which was pointed out in earlier studies (e.g. Stephens et al. 2015). Hemispherical albedo trends were analyzed by Datseries and Stevens (2021). Therefore, most of their results are reproduction of the results of earlier studies. Among three main results discussed in the discussion and summary session, only 3), which is the result of the comparison of different products, might be new. But the result is not essential in understanding the hemispherical albedo symmetry. In addition, as far as I know, AVHRR and ISCCP data products were not used in analyzing hemispherical albedo symmetry in earlier studies. Therefore, the motivation of the comparison is not clear. New knowledge added by this study is not significant enough to worth publication. However, because the extensive coverage of earlier studies, there might be a path forward. Because of the good coverage of earlier studies on this subject, I suggest converting this manuscript to a review paper.

   **Response:** We agree with the reviewer and are very appreciated for reviewer providing such helpful comments and suggestions. Indeed, our research results exhibit consistency and similarity with previous studies. In fact, the aim of this study is to interpret the characteristics of variations in hemispheric planetary albedo at finer temporal and spatial scales, thereby providing support for the investigation of hemispheric symmetry mechanisms. Additionally, we aim to systematically evaluate the applicability of existing long-term radiation data in hemispheric symmetry studies. We

acknowledge the reviewers' concern regarding the lack of emphasis on the manuscript's highlights, potentially causing confusion about its novelty. Following the suggestion from reviewer, therefore, we reorganize the manuscript, remove redundant content, and focus on addressing two key issues: (1) the characteristics of hemispheric reflected solar radiation variations at different latitudes and months, and (2) the ability of four datasets to accurately reproduce hemispheric differences and symmetry of reflected solar radiation observed by CERES.

**Specific Responses:**

1. Line 69-70 The location of ITCZ shifting with season is known before 2007.

   **Response:** Thanks for your comment! Indeed, as early as 1993, Waliser and Gautier (1993) systematically investigated seasonal variations in the intensity and location of the ITCZ. Hu et al. (2007) further noted that the seasonal shift in the global average ITCZ is not smooth, but jumps from the winter hemisphere to the summer hemisphere. The relevant reference is added in the revised manuscript: "For example, the Intertropical Convergence Zone (ITCZ) plays an important role in regulating cloudiness in the 10°S-10°N region, with its location and intensity varying seasonally (Waliser and Gautier, 1993; Hu et al., 2007)."

   Please see the Line: 69-71.

2. Line 70-71 There are many places that sentences are either awkward or do not make sense. In addition to this sentence, the sentence on line 84-85, line 290 "contribution rate", line 294 "molecular part of Eq. (14), line 360 (10a)-1 awkward units, for example.

   **Response:** We very thank reviewer for providing detailed comments and suggestions. We have carefully reviewed the manuscript and made the necessary revisions to address the issues you raised. Specifically, we have rephrased these sentences in the revised manuscript.

   Please see the Line: 73-76, 96-98, 296, 299-301, 357.

3. Line 80 the authors used "oblique pressure activity" several places in the manuscript. I think what they mean is mid-latitude baroclinic low pressure systems or synoptic systems, but I am not sure.

   **Response:** We are sorry to confuse the reviewer by the term "oblique pressure activity". It includes the mid-latitude baroclinic synoptic systems. For clarity, we correct the term and statement in the

revised manuscript: "In addition, recent studies have emphasized the impact of the distinct land-sea distribution between hemispheres, which leads to enhanced baroclinic activities at mid-latitudes in the SH, resulting in an increase in baroclinic synoptic systems (Hadas et al., 2023)."

Please see the Line: 82-85.

4. Line 88-90. If the authors are telling that aerosols increase deep convective clouds, could you cite papers?

Response: Thanks for your comments. Both model simulations (Wang et al., 2014) and satellite observations (Zhang et al., 2007) have pointed out that aerosols, as cloud condensation nuclei, act on the microphysics and dynamics of deep convective clouds, altering the cloud structure, elevating the cloud top heights, and inducing the development of deep convective clouds. The relevant references are added in the revised manuscript: "This, in turn, increases the amount of deep convective clouds due to the indirect effects of aerosols (Zhang et al., 2007; Wang et al., 2014). The increased deep convective clouds can strengthen the storm track in the Pacific Ocean and increase the contribution of the cloud component (Wang et al., 2014)."

Please see the Line: 101-103.

5. Line 97-98. I do not think that available data limit studying hemispherical albedo symmetry. The authors might mean studying how the symmetry changes with time?

Response: We are sorry that our expression has caused confusion to the reviewer. Yes, it means the available data limit the study of hemispherical albedo symmetry changes with time. In the revised manuscript, we correct the inappropriate statements: "However, the CERES observational record is relatively limited (2000-present), we cannot determine how hemispheric symmetry changes over time." (See the Line: 117-118).

6. Line 139 Why do the authors mention filtered radiance here?

Response: We are sorry to mislead the reviewer. Filtered radiation is mentioned here to explain the process of generating monthly regional TOA radiative fluxes from the original CERES measurements. We correct the inappropriate statements in the revised manuscript: "The radiance received by the CERES instrument is first converted from digital counts to calibrated "filtered"

radiances. This is then converted to unfiltered radiances to correct for imperfections in the spectral response of the instrument (Loeb et al., 2001), and then transformed into TOA instantaneous radiative fluxes using an empirical angular distribution model (Su et al., 2015). Instantaneous fluxes are converted to daily-averaged fluxes using sun-angle dependent diurnal albedo models (Loeb et al., 2018)."

Please see the Line: 160-165.

7. Section 2.2.1 is largely the reproduction of earlier study.

   **Response:** Thanks for your comments. In section 2.2.1, we have adopted the planetary albedo decomposition method proposed by Stephens et al. (2015) and therefore most of the formulas are derived from previous research. We retained this section for the purpose of enhancing clarity and providing a comprehensive understanding for the readers. This decision allows readers to understand the method more effectively within the context of our study.

8. Equation (12) Weighting by clear and cloud fraction is missing in the equation.

   **Response:** Thanks for your comments. The cloud contribution $F_{cloud}^{\uparrow}$ in Equation (12) is the difference between the all-sky atmospheric contribution $F_{atm}^{\uparrow}$ and the clear-sky atmospheric contribution $F_{atm,clear}^{\uparrow}$ (see equation (13)). Its definition is similar to the shortwave radiative effect of clouds, and thus not need to be weighted by clear and cloud fractions.

9. Equation (14) area weighted mean is probably sufficient instead of introducing the equation.

   **Response:** Thanks for your suggestion. In the revised manuscript, we replaced the regional weighted average method with the geodetic weighting method as recommended by the CERES website. This updated approach takes the Earth's oblate spheroid shape into account. Consequently, we have maintained these formulas and adjusted their introduction accordingly. The following is the revised text:

   "In calculating regional averages radiative flux, the study employs a geodesic weighting method consistent with the official CERES product. This method assumes Earth's oblate spheroid shape and takes into account the annual cycle of the Earth's declination angle and the sun-Earth distance (details about the method can be found in the website:

"https://ceres.larc.nasa.gov/documents/GZWdata/zone_weights.f"). The regional averaged TOA RSR $F_k$ is spatially aggregated using the following calculation formula:

$$F_k = \frac{\sum_{i=1}^{N_k} W_{ki} \cdot F_{ki}}{\sum_{i=1}^{N_k} W_{ki}} \tag{14}$$

Here, $N_k$ is the number of grid samples in region k, and $F_{ki}$ is the RSR flux corresponding to grid i in the region k. Moreover, $W_{ki}$ is the geodetic zonal weight for the grid i, which can be obtained from "https://ceres.larc.nasa.gov/documents/GZWdata/zone_weights_lou.txt". Regional averages for other variables are calculated according to the similar weighting equation."

Please see Line 283-293 for more details.

10. Equation (20). Generally, we do not put variables with different units in an equation. Correlation coefficient is a non-dimensional number while mean and RMS have units.

   **Response:** We very thank reviewer for providing detailed comments and suggestions. in the revised manuscript, the metrics (CC, AE, RMSE) are normalized to non-dimensional metrics (NCC, NAE, NRMSE), which are between 0 and 1, using the normalization formula following Chen et al. (2024) as:

$$NS_a = \frac{S_a - \min(S)}{\max(S) - \min(S)} \tag{20}$$

   Where S indicates the metric (CC, AE, and RMSE). Here, a=0, 1, …, m, "0" indicates the observed data, and m is the total number of model data used for comparison. Please note that Eq. (20) is a newly added equation in revised manuscript, and the Eq. (20) in the original manuscript is now renumbered as Eq. (21). After correcting this statistical error, we redraw the Fig. 5 in Section 3.2 in the revised manuscript (see the Fig. R1). The Fig. R1 (same as the Fig. 5a in the revised manuscript) indicates that the performance of ERA5 is better than the ISCCP in reproducing CERES observed hemispheric difference (NH-SH) of total RSR compared with our previous results. Please see Line 333-338 and 616-636 for more details.

**(a)CCHZ-DISO system with 3-dimension**

AVHRR(0.95,0.20,0.21) DISO1=0.29
ISCCP(0.58,0.41,0.42) DISO2=0.72
MERRA-2(0.00,1.00,1.00) DISO3=1.73
ERA5(0.76,0.48,0.48) DISO4=0.71

**Figure R1 (same as the Fig. 5a in the revised manuscript): CCHZ-DISO system with 3-dimension for hemispheric difference of annual-average total RSR between NH and SH. The coordinate axis consists of three statistical indicators, normalized correlation coefficient (NCC) for x-axis, normalized absolute error (NAE) for y-axis, and normalized root mean square error (NRMSE) for z-axis. OBS indicates observations, here referred to as CERES EBAF.**

11. Figures are generally too small to see the details.

   **Response:** Thanks for your comments. We have enlarged the text in all figures in the revised manuscript.

**Reference:**

Chen, F., Wang, D., Zhang, Y., Zhou, Y., and Chen, C.: Intercomparisons and Evaluations of Satellite-Derived Arctic Sea Ice Thickness Products, Remote Sensing, 16, 508, https://doi.org/10.3390/rs16030508, 2024.

Hadas, O., Datseris, G., Blanco, J., Bony, S., Caballero, R., Stevens, B., and Kaspi, Y.: The role of baroclinic activity in controlling Earth's albedo in the present and future climates, Proceedings of the National Academy of Sciences, 120, e2208778120, https://doi.org/10.1073/pnas.2208778120, 2023.

Hu, Y., Li, D., and Liu, J.: Abrupt seasonal variation of the ITCZ and the Hadley circulation, Geophysical research letters, 34, https://doi.org/10.1029/2007GL030950, 2007, 2007.

Loeb, N. G., Priestley, K. J., Kratz, D. P., Geier, E. B., Green, R. N., Wielicki, B. A., Hinton, P. O. R., and Nolan, S. K.: Determination of unfiltered radiances from the Clouds and the Earth's Radiant Energy System instrument, Journal of Applied Meteorology, 40, 822-835, https://doi.org/10.1175/1520-0450(2001)040<0822:DOURFT>2.0.CO;2, 2001.

Loeb, N. G., Doelling, D. R., Wang, H., Su, W., Nguyen, C., Corbett, J. G., Liang, L., Mitrescu, C., Rose, F. G., and Kato, S.: Clouds and the earth's radiant energy system (CERES) energy balanced and filled (EBAF) top-of-atmosphere (TOA) edition-4.0 data product, Journal of Climate, 31, 895-918, https://doi.org/10.1175/JCLI-D-17-0208.1, 2018.

Stephens, G. L., O'Brien, D., Webster, P. J., Pilewski, P., Kato, S., and Li, J.-l.: The albedo of Earth, Reviews of Geophysics, 53, 141-163, https://doi.org/10.1002/2014rg000449, 2015.

Su, W., Corbett, J., Eitzen, Z., and Liang, L.: Next-generation angular distribution models for top-of-atmosphere radiative flux calculation from CERES instruments: Methodology, Atmospheric Measurement Techniques, 8, 611-632, https://doi.org/10.5194/amt-8-611-2015, 2015.

Waliser, D. E. and Gautier, C.: A satellite-derived climatology of the ITCZ, Journal of climate, 6, 2162-2174, https://doi.org/10.1175/1520-0442(1993)006<2162:ASDCOT>2.0.CO;2, 1993.

Wang, Y., Wang, M., Zhang, R., Ghan, S. J., Lin, Y., Hu, J., Pan, B., Levy, M., Jiang, J. H., and Molina, M. J.: Assessing the effects of anthropogenic aerosols on Pacific storm track using a multiscale global climate model, Proceedings of the National Academy of Sciences, 111, 6894-6899, https://doi.org/10.1073/pnas.1403364111, 2014.

Zhang, R., Li, G., Fan, J., Wu, D. L., and Molina, M. J.: Intensification of Pacific storm track linked to Asian pollution, Proceedings of the National Academy of Sciences, 104, 5295-5299, https://doi.org/10.1073/pnas.0700618104, 2007.

---

## Author Comment (AC2)

**Response to Reviewer #2's Comments:**

Ruixue Li et al. (Author)

**We are very grateful for the Reviewer #2' constructive comments and suggestions, which help us improve this paper significantly. Based on the comments from reviewers, we reorganize the abstract, introduction, datasets and methods, results and conclusion sections, and add some interpretations in each section in order to enhance the manuscript's continuity and depth of analysis. In addition, some superfluous information in each section is deleted. Based on the comments and suggestions, we also correct inappropriate or unclear descriptions in the manuscript.**

**Important revision includes:**

(1)  Section 1, Introduction, is rewritten to enhance clarity regarding the significance and innovations of the manuscript. Superfluous information is removed, and specifically, we highlight the reasons for studying the changes in reflected solar radiation (RSR) at finer spatiotemporal scales and the importance of assessing the applicability of different radiation datasets with longer record in hemispheric symmetry studies. More details are described in Line 65-138. The abstract is also further reorganized to summarize the main findings of this paper.

(2)  In Section 2, Datasets and Methodology, we update the CERES data to the latest version (CERES-EBAF 4.2) and correct some bugs in the data processing, and revise the corresponding descriptions.

(3)  We agree the reviewer's comments. In Section 3.1, we acknowledge that there is a little bit overlap between Fig. 1 and previous studies. However, the focus of this section is on analyzing changes in RSR and its components at the latitude and monthly scales. We have reorganized Section 3.1 and added an expanded discussion on hemispheric RSR trends, highlighting the contributions of finer spatiotemporal scales to these trends. Additionally, we have analyzed the changes in the contribution rates of different components to the RSR.

(4)  The content discussing the contribution of different factors to latitudinal zones in extreme years (Section 3.2) is removed. The adjustment allows us to maintain a more focused and streamlined discussion, centering on the primary objectives and findings of the study.

(5)  We substantially expand the original Section 3.3 (now revised as Section 3.2) to systematically quantify the performance of four radiation datasets in reproducing hemispheric differences and symmetry

of RSR observed by CERES at hemispheric and finer temporal-spatial scales. This expansion ensures a smoother and tighter connection with Section 3.1.

(6) In the Section 4, Discussion and Summary, we revise the main conclusions of this study and delete some superfluous information.

(7) In each section, we also add some interpretations about the comments from reviewers

**Please see our point-by-point reply to comments. All revisions are shown in revised manuscript by using track changes.**

**General responses:**

1. In Section 3.3, rather than simply comparing with CERES for the asymmetry issue in modeled PA, the analysis should be integrated more cohesively with key findings from previous sections. This integration is essential for establishing a stronger motivation and relevance for the comparison. For instance, it would be valuable to assess whether the model data captures the interannual anomaly of the contribution rate of different components to total reflected radiation, thus linking the analysis with earlier sections and enhancing the manuscript's continuity and depth of analysis.

   **Response:** We very thank reviewer for providing detailed and constructive comments and suggestions. Based on the comments and suggestions from two Reviewers, in the revised manuscript, we delete the Section 3.2 and substantially expanded the original Section 3.3 (now revised as Section 3.2) to systematically quantify the performance of four radiation datasets in reproducing hemispheric differences and symmetry of reflected solar radiation observed by CERES at hemispheric and finer temporal-spatial scales. Please see Section 3.2.

2. Simulated Snow uncertainty has been suggested to introduce substantial bias of surface albedo among reanalysis data, especially at mid and high latitudes [1]. Are there any influences for the TOA asymmetry issue in simulations?

   **Response:** Thanks for your comments. Surface snow cover significantly affects surface albedo, especially at middle and high latitudes. In the process of calculating reflected solar radiation, surface parameters, including snow products, are introduced into the radiative transfer model, which may

introduce bias. The inaccuracy in snow cover products can introduce some bias in the top-of-atmosphere radiative fluxes and potentially influence hemispheric differences of reflected solar radiation to some extent. The relevant reference and discussion are added in the revised manuscript:"Moreover, MERRA-2 significantly underestimates surface components in Antarctica during melting season (November to January), which could be due to biases in the input snow products that introduce significant uncertainties in surface albedo (Jia et al., 2022)."

Please see the Line: 755-757.

3. Line 546: despite including various driving factors (e.g., NDVI, snow cover) for anomaly attribution, their corresponding radiative forcings differ significantly. Consequently, even if two factors exhibit similar anomaly magnitudes in a given year, the importance of NDVI may not be comparable to snow cover changes, rendering the anomaly analysis less meaningful. Therefore, I recommend converting the anomaly analysis to a corresponding radiative forcing analysis to better capture the relative importance of different factors in driving changes in radiative forcing over time.

**Response:** We very thank reviewer for providing detailed comments and suggestions. We agree with the reviewer. Indeed, even if two factors exhibit similar magnitudes of anomalies in a given year, their relative importance may not be comparable. As stated in original manuscript: "the radiation contributions from different latitudinal zones exhibit varying sensitivities to changes in different factors, resulting in different magnitudes of response." Therefore, in previous analyses, the relative anomalies of factors serve merely as a reference for identifying and analyzing anomaly events. Based on the comments and suggestions from two Reviewers, we reorganize the manuscript and only focus on two key issues: (1) analyze the characteristics of the variations in hemispheric reflected solar radiation at a latitude- and month-based perspective; and (2) systematically quantify the performance of four radiation datasets in reproducing hemispheric differences and symmetry of reflected solar radiation observed by CERES. In the revised manuscript, we reorganize the section of results in order to make the manuscript clearer. Thus, the content discussing the contribution of different factors to latitudinal zones in extreme years (Section 3.2) is already removed and some ambiguous descriptions are also corrected in the revised manuscript.

4. The difference of r in Eqs. 4 and 5: r in Eq.4 represents blue-sky reflectance, where the solar beam reflects from the surface, whereas the r in eq5 is black-sky reflectance and the incoming radiation is from space. Does this difference in physics have any impact, particularly at high latitudes where the SZA is large?

**Response:** We very thank reviewer for providing detailed comments and suggestions. We fully understand the reviewer's concerns and apologize for any confusion caused by our unclear expression. Black-sky albedo is the intrinsic albedo of the surface when atmospheric diffuse is ignored. Blue sky albedo is the actual surface albedo. However, r in Eq. 4 and Eq. 5 represents atmospheric intrinsic reflectivity, which is independent of the surface albedo. We correct the inappropriate statements in the revised manuscript: "Here, r and t represent atmospheric intrinsic reflectivity (that is, PA purely contributed by the atmosphere) and atmospheric transmittance, respectively."

Please see the Line: 260-261.

**Specific Responses:**

1. Suggest introducing parameters with 0 in Eq. 20

**Response:** Thanks for your suggestions. We have added the introduction for parameters with 0 in Eq. 21 (same as the Eq. 20 in the original manuscript) in the revised manuscript:

$$DISO_i^{xj} = \sqrt{(CC_i - CC_0)^2 + (NAE_i - NAE_0)^2 + (NRMSE_i - NRMSE_0)^2} \qquad (21)$$

Where $i$ and $xj$ represent the $i$th model and $j$th variable. The subscript "0" in Eq. 21 represents statistical parameters of variable $xj$ from observation data (here refers to CERES EBAF).

Please see Line: 340-341.

2. Figure 2a: I suggest changing the color bar because the conventional association of 'blue & red' typically implies negative and positive directions, whereas the result here is uni-directional.

**Response:** We very thank reviewer for providing detailed comments and suggestions. We have modified the inappropriate color bar of figures in the revised manuscript. Additionally, based on the comments and suggestions from two Reviewers, we reorganize the Section 3.1 to only focus on analyzing the characteristics of the variations in hemispheric reflected solar radiation at a latitude- and month-based perspective, rather than on the contribution of different latitudinal zones to

hemispheric total reflected radiation at TOA. As a result, we replace the related content regarding the "contribution of different latitudinal zones to hemispheric total reflected radiation at TOA from 2001 to 2021 and the corresponding components (Figure 2a-d in the original manuscript)" with "reflected solar radiation of different latitudinal zones at TOA from 2001 to 2021 and the corresponding components (Figure 3a-d in the revised manuscript, please see the Figure R1)". Although the variables shown here are different, they essentially reflect the same information.

[Figure]

**Figure R1 (same as the Fig. 3a-d in the revised manuscript): Annual averages from 2001 to 2021 of (a)total reflected solar radiation flux and its (b) clear-sky atmospheric component, (c) surface component, and (d) cloud component at different latitudinal zones.**

3. Line 437: The dominant component in the NH?

   **Response:** It is corrected in the revised manuscript: "The higher RSR from the 0°-40° latitude zones in the NH stems from the higher cloud component from the equator to 10° and the combined effect of clear-sky atmospheric and surface components in the 10°-40°."

   Please see the Line: 469-471.

4. Line 438: 0°-70°? Why does it have some overlaps with the following ones?

   **Response:** Thanks for your comments. We correct the inappropriate statements in the revised manuscript: "The higher RSR from the 0°-40° latitude zones in the NH stems from the higher cloud component from the equator to 10° and the combined effect of clear-sky atmospheric and surface components in the 10°-40°."

   Please see the Line: 469-471.

5. The figures should be enlarged.

   **Response:** Thanks for your comments. We have enlarged the text in all figures in the revised manuscript.

**Reference:**

Jia, A., Wang, D., Liang, S., Peng, J., and Yu, Y.: Global daily actual and snow-free blue-sky land surface albedo climatology from 20-year MODIS products, Journal of Geophysical Research: Atmospheres, 127, e2021JD035987, https://doi.org/10.1029/2021JD035987, 2022.

---

## Referee Report (RR1)

**Review of:**
**"Understanding the variation of Reflected Solar Radiation: A Latitude- and month-based Perspective", Li et al.**

The authors conduct a statistical analysis of the evolution of the hemispheric albedo symmetry in the observational record through looking at the components that comprise it, the annual cycle, and where in the world/which component of albedo/when in the year trends are occurring. The analysis aids in studies of the hemispheric albedo symmetry by breaking down trends in components of the albedo symmetry into regions, latitudes, and months, which would point the community towards details around its changing nature. Comparisons between data sets and reanalyses would help advise the community in using the right tools to understand the hemispheric albedo symmetry. The description of the data sets and methods is well-written and thorough.

I would recommend the manuscript to be accepted after some minor revisions. My two major comments below address things that I feel need to be addressed in order for the analysis to be more robust, especially with using albedo as fractional terms in addition to the energetic terms presented. The rest of my comments are aimed at smaller errors or points of confusion.

More generally: although I understood and the paper is generally structured well, I had difficulty with the language and grammar, and I suggest that a language service be used in revising the manuscript. I would caution the authors to use the word "significant" carefully as it is generally used to mean statistical significance, but I have difficulty understanding which is meant throughout the manuscript.

Sincerely,

Aiden Jönsson

**Major Comments**

Since CERES is itself an observational data set assimilating multiple sources of satellite-based observations with its own weaknesses, it should not be considered truth. I recognize that the albedo symmetry is primarily a feature able to be studied through CERES, but statements of it being more real/true than other data sets by nature could be relaxed throughout the manuscript.

L389 and the following paragraph: There is a reduction from spring to winter in both hemispheres, but since these are in energetic units, it could be good to show Figure 2 using albedo as fractional terms in order to remove the seasonal insolation cycle. Please consider replacing Figure 2 with that, or including it in the supplement. I'm not sure how these will affect the results, but I think a stronger decreasing trend in energetic terms during summer can be expected.

L945-957: The authors calculate a trend in CERES Earth energy imbalance (EEI) and introduce the result in the Discussion and Conclusions section. Accurately observing and calculating the EEI is not an easy task and should probably not be done in this way; it would probably be best to refer to previous studies on this, and if the authors wish to include it in the results, they should do so before this section.

**Minor comments**

Title: This paper has more to do with trends than it does variation/variability; would it be more fitting to call it "Understanding trends in Reflected Solar Radiation: …"?

L8: It is unclear what "hemispheric variations" may mean for signals in the hemispheric symmetry's development or trends.

L11: It is unclear what is being reproduced: reproducing the hemispheric symmetry would to me indicate a modeling study, but I believe the authors intend to reproduce analyses of the hemispheric symmetry in other observational records. It would help to clarify what is being done here.

L13: Here and elsewhere: when saying "decreasing trends" or so, please specify what is decreasing, such as explicitly stating "trends in decreasing RSR".

L21-22 and elsewhere (e.g. L26): "Reproducing hemispheric symmetry": This makes it seem as if the symmetry primarily studied in CERES is an absolute truth, and other data sets would "fail" if they do not have symmetry. If I am understanding correctly that this sentence means that AVHRR *exhibits* hemispheric symmetry, then perhaps it would help to phrase it so.

L32-33: It would help the flow to use percentage or fractional terms consistently (e.g. 5% and 1%, or 0.05 and 0.01).

L53: Please introduce CMIP/its full name before defining it as an abbreviation.

L89: Could you please clarify what "longer storm tracks" implies – are they longer in the temporal dimension, for example?

L92: Specify that forest fires occur during summer and autumn, not volcanic eruptions.

L100: Please expand or clarify "aerosol effects"; long-range aerosol transport can affect both AOD at range, but may also affect clouds.

L102: The deep convective region and the storm track are quite remote to one another, and it isn't obvious how one affects the other. Please expand and clarify what is meant by this effect, and how it occurs.

L117: They did find that model projections would suggest a symmetry breaking with warming, but I think a more open interpretation of the results – that "will be disrupted" can be relaxed to "may be disrupted" – could better reflect their conclusions.

L117: Connecting word ("… *because* the CERES record …") missing.

L126: The term AVHRR is not introduced before using it.

L129: MERRA-2 and ERA5 should also be introduced and cited.

L276: Decomposing into cloudy and clear-sky atmospheric flux contributions was already done in Stephens et al (2015).

L372: Citing Diamond et al (2022) would be helpful here, since the topic is the clear-sky asymmetry.

L419: Suggest: reduce → reducing

L422: Suggest: link → relate/be related

L429: "This might be attributed to decreasing cloud cover": I am not sure what is meant by this statement.

L435: "where is" → "where it is"

L452: Compensate for RSR in what way?

L461: "where has" → "where there is"

L465: The citation George and Bjorn, 2021 needs to be fixed (Datseris and Stevens, 2021).

L473: Suggest "more radiation" → "more reflected radiation"

L475: Missing a connecting word or phrase between "the NH as a whole" and "slightly higher"

L481-485: These repeat L471-473, suggest combining to shorten.

L486: The citation should be Bender et al., 2017; the author order needs to be fixed.

L490-492: It is not clear what this sentence offers in terms of conclusions; it needs to be reformulated more specifically regarding hemispheric differences by latitude to be a helpful summarizing statement.

L493-503: This paragraph would address an essential part of the analysis and provide some good insights, but I find it hard to understand. The annual cycle could be removed to aid this, and I suggest the use of albedo as fractional terms rather than reflected radiation in W m$^{-2}$ here to help.

L504: Is it supposed to be "decadal/secular" trend rather than "interannual trend"?

L509: Not sure if "disparate" is used correctly.

L537-541: There is also the effect of cloud reductions, which "unmasks" some of the clear-sky component. Thus the masking effect may explain at least some of the clear-sky increasing trends over the SH midlatitudes. This may also help the following two paragraphs' analysis in contribution rates' trends.

L583-584: This reflects previous results as well, such as Sledd and L'ecuyer (2021a) and Sledd and L'ecuyer (2021b).

Figure 5a: It may be good to label the "OBS" origin in the plot as "CERES" instead, since there are multiple observational data sets here, and it could be good to not regard CERES as absolute truth.

L608-609: These sentences should be written in complete form.

L609: It could help to make the figure stand alone better if DISO is defined in the caption.

L621: Suggest "Note that the good performance of dataset" → "Note that a data set may perform well because …"

L654-655: I understand what this sentence is saying, but the grammar makes it difficult to read.

L681: What is it that it exhibits the poorest of?

L774: "Length of year" is an odd choice of words as it sounds like the year length is varying. It would be better to say "Length of time series [years]" instead.

L782: Citation needs to be fixed as in L465.

L794: The quantification of the symmetry has been discussed in various ways throughout the literature; please clarify what it is that has not been shown before.

L806: "More than two years" is a very imprecise description, please be specific.

L812: What does "regional average" mean here? Across which spatial scale?

Figure 8d: There is a clear meridionally dependent bias with too much clear-sky atmospheric component in the tropics and too little at the poles in AVHRR. Is there an explanation for this, or can the authors comment on this? Perhaps water vapor affecting transmissivity can play a role here, since the $t$ term in the transfer model can be reduced by increasing absorption of water vapor.

L875: "Unreal" is a loaded word to be used in scientific writing. Here and elsewhere I would suggest using phrases such as "not seen in CERES" rather than unreal or spurious.

L877: Clouds are not parameterized in observational data sets or radiative transfer models, or at least not in the same way that they are in reanalysis forecast models. I would suggest clarifying this earlier in the manuscript.

L889: What is meant by "powerful"? Perhaps "robust"?

L891-893: There have not been many compensation mechanisms suggested as of yet.

L908-910: This sentence makes it sound like the connection between the PDO and the North Atlantic clouds is trivial and well-documented, although I do not think this is what the authors meant to convey.

L917-919: What is meant by the "the contribution rate from 30-40° N to the hemisphere"?

L921: The term "outstanding" is a bit too extreme here.

L955: There is a more recent IPCC Assessment Report that can be cited to better reflect the current state of understanding.

---

## Author Response (AR2)

**Response to Reviewer #3's Comments:**

Ruixue Li et al. (Author)

**We are very grateful for the Reviewer #3' detailed and constructive comments and suggestions, which help us improve this paper greatly. On this basis, we correct inappropriate or unclear descriptions in the manuscript. We have also included additional substantial discussions to enrich and complete the content of the manuscript.**

**Please see our point-by-point reply to comments. All revisions are shown in revised manuscript by using track changes.**

**General responses:**

1. Although I understood and the paper is generally structured well, I had difficulty with the language and grammar, and I suggest that a language service be used in revising the manuscript. I would caution the authors to use the word "significant" carefully as it is generally used to mean statistical significance, but I have difficulty understanding which is meant throughout the manuscript.

   **Response:** Thank you very much for providing detailed comments and suggestions. The language is checked in the revised manuscript.

**Major responses:**

1. Since CERES is itself an observational data set assimilating multiple sources of satellite-based observations with its own weaknesses, it should not be considered truth. I recognize that the albedo symmetry is primarily a feature able to be studied through CERES, but statements of it being more real/true than other data sets by nature could be relaxed throughout the manuscript.

   **Response:** We very thank you for providing detailed and constructive comments and suggestions. We agree with the reviewer. Indeed, CERES itself suffers from uncertainties such as calibration errors (Loeb et al., 2018), and thus the accuracy of its records is also limited. We are sorry for the inappropriate statements made in the previous version of the manuscript. In the revised manuscript, we have adjusted the language to ensure a balanced perspective of the strengths and limitations of the other datasets versus CERES.

2. L389 and the following paragraph: There is a reduction from spring to winter in both hemispheres, but since these are in energetic units, it could be good to show Figure 2 using albedo as fractional terms in order to remove the seasonal insolation cycle. Please consider replacing Figure 2 with that, or including it in the supplement. I'm not sure how these will affect the results, but I think a stronger decreasing trend in energetic terms during summer can be expected.

**Response:** We very thank reviewer for providing detailed comments and suggestions. In order to remove the seasonal insolation cycle, the hemispheric average monthly trends of incident solar radiation at the top of the atmosphere (Figure S2 in the revised manuscript, please see the Figure R1) and planetary albedo (PA) and its components (Figure S3 in the revised manuscript, please see the Figure R2) are given in the revised manuscript and supplement. The detailed discussion is added in the revised manuscript: "Considering the potential impact of the annual cycle of incident solar radiation, we have also provided the hemispheric average monthly trends of TOA incident solar radiation (Fig. S2) and PA (Fig. S3). The results show that there are no significant long-term trends in the incident solar radiation among all months. However, after removing the effects of seasonal variations in incident solar radiation, the monthly trends in PA for both hemispheres exhibit distinct differences from the total RSR. Under these conditions, the decreasing trends in PA during winter are comparable to those observed during summer in both hemispheres. This suggests that the weaker decreasing trend of RSR in autumn and winter compared to that in spring and summer may be affected by the reduced incident solar radiation during these seasons.". Please see the Line: 397-404.

[Figure]

Figure R1 (same as the Fig. S2 in the revised manuscript): The hemispheric averaged trends in incident solar radiation at the top of atmosphere in the (a) NH and (b) SH for different month from 2001-2021.

[Figure]

**Figure R2 (same as the Fig. S3 in the revised manuscript): The hemispheric averaged trends in planetary albedo and its components in the (a) NH and (b) SH for different month from 2001-2021. Pink, yellow and blue bars indicate trends in the clear-sky atmospheric component, surface component and cloud component, respectively. The brown line indicates the trend of planetary albedo. Dots of different colours indicate that the hemispheric averaged trend of the corresponding variable is significant at the 95% confidence level.**

3.  L945-957: The authors calculate a trend in CERES Earth energy imbalance (EEI) and introduce the result in the Discussion and Conclusions section. Accurately observing and calculating the EEI is not an easy task and should probably not be done in this way; it would probably be best to refer to previous studies on this, and if the authors wish to include it in the results, they should do so before this section.

    **Response:** We very thank you for providing detailed comments and suggestions. We agree with the reviewer. In fact, there is currently no direct method to measure the magnitude of EEI (Trenberth and Fasullo, 2010). The uncertainty in the net radiation flux at the top of the atmosphere provided by CERES may very differ in magnitude from the size of EEI, and therefore cannot accurately represent the absolute magnitude of EEI. Therefore, we did not attempt to quantify EEI. However, due to the stability of CERES observations, its global data can provide support for the study of the relative changes in EEI over time (Trenberth et al., 2014; Loeb et al., 2021; Trenberth and Cheng, 2022). Based on the reviewer's suggestion, we have removed our calculated results and cited the findings of Loeb et al. (2022) in the revised manuscript: "In addition, previous study (Loeb et al., 2022) pointed out that a significant increasing trend of LW radiation emitted to space is found in the NH, while no significant trend is observed in the SH." Please see the Line: 990-992.

**Specific Responses:**

1. Title: This paper has more to do with trends than it does variation/variability; would it be more fitting to call it "Understanding trends in Reflected Solar Radiation: …"?

   **Response:** Thank you for your suggestion, the title has been changed to "Understanding the trends in Reflected Solar Radiation: A Latitude- and month-based Perspective" in the revised manuscript.

2. L8: It is unclear what "hemispheric variations" may mean for signals in the hemispheric symmetry's development or trends.

   **Response:** We are sorry that our expression has caused confusion to the reviewer. In the revised manuscript, we correct the inappropriate statements: "Averaging reflected solar radiation (RSR) over the whole year/hemisphere may mask the inter-month/region-specific signals, limiting the investigation of spatiotemporal mechanisms and hemispheric symmetry projections." (See the Line: 8-10)

3. L11: It is unclear what is being reproduced: reproducing the hemispheric symmetry would to me indicate a modeling study, but I believe the authors intend to reproduce analyses of the hemispheric symmetry in other observational records. It would help to clarify what is being done here.

   **Response:** Thanks for your comments. The term "reproduce" may not accurately reflect the intention of the study. We have rephrased this term in the revised manuscript: "The study also explores whether longer-record radiation datasets can exhibit hemispheric symmetry of RSR to understand its temporal changes." (See the Line: 11-12)

4. L13: Here and elsewhere: when saying "decreasing trends" or so, please specify what is decreasing, such as explicitly stating "trends in decreasing RSR".

   **Response:** We very thank reviewer for providing detailed comments and suggestions. We have carefully checked the language and made the necessary revisions in the revised manuscript.

5. L21-22 and elsewhere (e.g. L26): "Reproducing hemispheric symmetry": This makes it seem as if the symmetry primarily studied in CERES is an absolute truth, and other data sets would "fail" if they do not have symmetry. If I am understanding correctly that this sentence means that AVHRR exhibits hemispheric symmetry, then perhaps it would help to phrase it so.

   **Response:** Thank you very much for your detailed suggestion. In fact, what we intend to express is whether other datasets can also exhibit the hemispheric symmetry in reflected solar radiation as

observed by CERES. We are sorry that the term "reproduce" may be misleading and we have corrected the inappropriate expression in the revised manuscript.

6. L32-33: It would help the flow to use percentage or fractional terms consistently (e.g. 5% and 1%, or 0.05 and 0.01).

**Response:** Thanks for your detailed comments. For clarity, we correct the term and use percentages consistently in the revised manuscript: "Studies have shown that a 5% change in PA can lead to an average global temperature change of approximately 1K (North et al., 1981), while a 3% change in PA can have a radiative forcing effect equivalent to doubling the amount of carbon dioxide in the atmosphere". Please see the Line: 32-34.

7. L53: Please introduce CMIP/its full name before defining it as an abbreviation.

**Response:** Thanks to your comment, it is added in the revised manuscript. Please see the Line: 54.

8. L89: Could you please clarify what "longer storm tracks" implies – are they longer in the temporal dimension, for example?

**Response:** We are sorry to mislead the reviewer. It should actually be "stronger". Due to the marked weakening of the summer baroclinicity, the intensity of the storm tracks in winter is nearly three times than that in summer for Northern Hemisphere (Hadas et al., 2023). The expression is corrected and relevant reference is added in the revised manuscript. Please see the Line: 90.

9. L92: Specify that forest fires occur during summer and autumn, not volcanic eruptions.

**Response:** Thank you for your suggestion, it is corrected in the revised manuscript: "These events usually occur in certain regions and forest fires occur typically during the summer and autumn (Fan et al., 2023), but they have important impacts on the interannual hemispheric symmetry of RSR." Please see the Line: 93-95.

10. L100: Please expand or clarify "aerosol effects"; long-range aerosol transport can affect both AOD at range, but may also affect clouds.

**Response:** Thank you for your comment. it is clarified in the revised manuscript: "For example, anthropogenic emissions from Asia not only enhance the local clear-sky atmospheric component of RSR through direct aerosol effects but also increase aerosol optical thickness in the northwestern Pacific through long-range transport. The long-range aerosol transport can also affect clouds by elevating cloud condensation nuclei (CCN) levels through the indirect effects of aerosols, thereby increasing cloud droplet number concentration, liquid water content, and updraft velocity. And it

can increase the amount of deep convective clouds and lead to suppressed coalescence and warm rain but efficient mixed-phase precipitation (Zhang et al., 2007; Wang et al., 2014)." Please see the Line: 99-106.

11. L102: The deep convective region and the storm track are quite remote to one another, and it isn't obvious how one affects the other. Please expand and clarify what is meant by this effect, and how it occurs.

Response: We are sorry to confuse the reviewer. Both the deep convection region and the Pacific storm track are located in the North Pacific Ocean. The long-range transport of anthropogenic emissions from Asia markedly increases the aerosol optical depth over the northwestern Pacific, which in turn substantially increases cloud droplet number concentration and liquid water content, promoting the development of deep convective clouds and large-scale convection and precipitation, and consequently enhancing the Pacific storm tracks in Northern Hemisphere (area range is approximately 120˚E~120˚W, 30˚N~60˚N). It is clarified in the revised manuscript: "The long-range aerosol transport can also affect clouds by elevating cloud condensation nuclei (CCN) levels through the indirect effects of aerosols, thereby increasing cloud droplet number concentration, liquid water content, and updraft velocity. And it can increase the amount of deep convective clouds and lead to suppressed coalescence and warm rain but efficient mixed-phase precipitation (Zhang et al., 2007; Wang et al., 2014). The increased deep convective clouds and changed cloud microphysical processes over the northwestern Pacific can strengthen the NH storm track in the Pacific Ocean via large-scale enhanced convection and precipitation, thereby increase the contribution of the cloud component (Zhang et al., 2007; Wang et al., 2014)". Please see the Line: 101-109.

12. L117: They did find that model projections would suggest a symmetry breaking with warming, but I think a more open interpretation of the results – that "will be disrupted" can be relaxed to "may be disrupted" – could better reflect their conclusions.

Response: Thank you very much for providing detailed suggestions. It is rephrased in the revised manuscript. Please see the Line: 123.

13. L117: Connecting word ("... because the CERES record …") missing.

Response: We are sorry for missing this connecting word. It is added in the revised manuscript. Please see the Line: 123.

14. L126: The term AVHRR is not introduced before using it.

    **Response:** Thanks to your comment, it is added in the revised manuscript. Please see the Line: 133.

15. L129: MERRA-2 and ERA5 should also be introduced and cited.

    **Response:** Thanks for your comments. The full name and relevant references are added in the revised manuscript. Please see the Line: 136-138.

16. L276: Decomposing into cloudy and clear-sky atmospheric flux contributions was already done in Stephens et al (2015).

    **Response:** Thank you for your comment. The relevant reference is added in the manuscript. Please see the Line: 282.

17. L372: Citing Diamond et al (2022) would be helpful here, since the topic is the clear-sky asymmetry.

    **Response:** Thank you very much for your suggestions. The relevant reference is added in the manuscript. Please see the Line: 379.

18. L419: Suggest: reduce → reducing

    **Response:** Thanks for your detailed suggestion. It is corrected in the revised manuscript. Please see the Line: 433.

19. L422: Suggest: link → relate/be related

    **Response:** Thanks for your detailed suggestion. It is corrected in the revised manuscript. Please see the Line: 436.

20. L429: "This might be attributed to decreasing cloud cover": I am not sure what is meant by this statement.

    **Response:** We are sorry to confuse the reviewer. It means that the decreasing trend in reflected solar radiation in most months in the Southern Hemisphere is primarily driven by the decrease in cloud component from 0-60°S. This decrease in cloud component may be related to reductions in cloud cover in some areas, such as a decrease in low clouds over tropics and a decrease in total cloud cover over the Southern Ocean (mentioned later in the sentence). It is clarified in the revised manuscript: "This may be partly attributed to decreasing cloud cover in specific regions, such as the tropics and the Southern Ocean. On the one hand, the low cloud cover over tropics has decreased due to the increasing SST. On the other hand, multi-source satellite cloud climatological data consistently show a significant decreasing trend in total cloud cover over the Southern Ocean". Please see the Line: 442-445.

21. L435: "where is" → "where it is"

    **Response:** Thanks for your detailed suggestion. It is corrected in the revised manuscript. Please see the Line: 448.

22. L452: Compensate for RSR in what way?

    **Response:** We are sorry that our expression has caused confusion to the reviewer. It means the "compensate mechanism for hemispheric symmetry of RSR". Regarding the compensation mechanism for hemispheric symmetry of RSR, Voigt et al. (2014) has pointed out that the ITCZ maybe an important compensation mechanism for RSR hemispheric symmetry, which shifts the it toward the darker surface hemisphere. Since the relevant content was introduced in the Line 70-88, it will not be repeated here. In the revised manuscript, we correct the inappropriate statements: "While larger regional anomalies in RSR may offset each other when spatially and temporally averaged to calculate global RSR and its interannual variations, these anomalies play a crucial role in regional radiation budgets, subsequent climate change, and the identification of mechanisms that maintain or compensate for hemispheric symmetry of RSR".

23. L461: "where has" → "where there is"

    **Response:** Thanks for your detailed suggestion. It is corrected in the revised manuscript. Please see the Line: 475.

24. L465: The citation George and Bjorn, 2021 needs to be fixed (Datseris and Stevens, 2021).

    **Response:** We are sorry for the incorrect citation format used. It is corrected in the revised manuscript. Please see the Line: 480.

25. L473: Suggest "more radiation" → "more reflected radiation"

    **Response:** Thanks for your detailed suggestion. It is added in the revised manuscript. Please see the Line: 488.

26. L475: Missing a connecting word or phrase between "the NH as a whole" and "slightly higher"

    **Response:** Thanks for your detailed suggestion. It is added in the revised manuscript. Please see the Line: 490.

27. L481-485: These repeat L471-473, suggest combining to shorten.

    **Response:** Thank you very much for your suggestion. It is reorganized in the revised manuscript: "At the high latitudes of 70°-80°, the SH shows larger surface reflections due to higher snow and

ice cover in the near polar regions. Between 50°-60°, cloud components in the SH reflect more solar radiation and reach maximum hemispheric differences". Please see the Line: 496-499.

28. L486: The citation should be Bender et al., 2017; the author order needs to be fixed.

    **Response:** We are sorry for the incorrect citation. It is corrected in the revised manuscript. Please see the Line: 500.

29. L490-492: It is not clear what this sentence offers in terms of conclusions; it needs to be reformulated more specifically regarding hemispheric differences by latitude to be a helpful summarizing statement.

    **Response:** Thanks to your helpful suggestions! We have reorganized this paragraph to make the conclusions clearer and more informative. "Based on the above analyses, we can find that the RSR and its components in the corresponding latitude zones of the two hemispheres are asymmetric. It is the offsetting of the differences in the different components across the latitudinal zones that leads to the minimal hemispheric differences in total RSR -- the cloud component in the mid-latitudes and the surface component in the high latitudes of the SH offset the clear-sky reflectance in the mid-low latitudes of the NH." Please see the Line: 504-509.

30. L493-503: This paragraph would address an essential part of the analysis and provide some good insights, but I find it hard to understand. The annual cycle could be removed to aid this, and I suggest the use of albedo as fractional terms rather than reflected radiation in W m$^{-2}$ here to help.

    **Response:** We very thank reviewer for providing important comments and suggestions. In this paragraph, the Figs. S6-S11 in the supplement are intended to provide further insights into the results of Figures 3e-h. Our focus is primarily on the hemispheric differences in the monthly variation of the total reflected solar radiation (RSR) in each latitudinal zone, rather than on the hemispheric differences in the monthly variation of the reflective capacity of each latitudinal zone. The annual cycle of the planetary albedo (PA) in different latitude zones and their hemispheric differences are shown in Fig. R3. Compared to the Figure R4 (same as the Fig. S6 in the revised manuscript), we can see that at high latitudes (70°-80°) during the winter, the hemispheric differences (NH-SH) of PA present a shift from positive to negative, but the hemispheric difference is almost zero in terms of RSR, as the incident solar radiation in both hemispheres is close to zero during the period (Fig. R5, same as the Fig. S6 in the revised manuscript). Moreover, the albedo

information does not capture the effect of offsetting the hemispheric differences in RSR by different months. Therefore, we have chosen not to replace the reflected solar radiation with planetary albedo. To enhance the clarity of our discussion, we have reorganized the paragraphs: "In addition, to clarity the variations and hemispheric differences of RSR at finer temporal scale, we further analyze the annual cycle of hemispheric differences of RSR across different latitudinal zones. Figure S6-S9 illustrate the annual cycle of RSR and its components in different latitudinal zones and their interhemispheric differences. It can be seen that the hemispheric differences of RSR in different latitudinal zones present obvious monthly variations, with the peak values in summer and winter. At middle-high latitudes, the annual cycles of the hemispheric differences of RSR and its components is relatively consistent with that of incident solar radiation (Fig. S10). However, the surface components in the 40°-60°N latitudinal zones exhibit enhanced reflectivity and interhemispheric differences in spring (Fig. S8), possibly influenced by surface albedo (Fig. S11). The annual cycle of hemispheric RSR differences is dominated by the cloud component at mid-low latitude and the surface component at high-latitude." Please see the Line: 510-520.

[Figure]

Figure R3: Annual cycle of the planetary albedo in different latitude zones and their hemispheric differences (NH-SH). The blue bars are for the NH, orange for the SH, corresponding to the left axis; the green line represents the inter-hemispheric difference, corresponding to the right axis, and the green shading indicates the difference spread in hemispheric difference for the corresponding month in the latitude zone during 2001-2021. The months are marked according to the NH, corresponding to the SH months of September, October, ..., January, February, ..., and August.

[Figure]

**Figure R4 (same as the Fig. S6 in the revised manuscript): Annual cycle of the total RSR in different latitude zones and their hemispheric differences (NH-SH). The blue bars are for the NH, orange for the SH, corresponding to the left axis; the green line represents the inter-hemispheric difference, corresponding to the right axis, and the green shading indicates the difference spread in hemispheric difference for the corresponding month in the latitude zone during 2001-2021. The months are marked according to the NH, corresponding to the SH months of September, October, ..., January, February, …, and August.**

[Figure]

**Figure R5 (same as the Fig. S10 in the revised manuscript): same as Fig. R4, but for incident solar radiation at the TOA.**

31. L504: Is it supposed to be "decadal/secular" trend rather than "interannual trend"?

    **Response:** Thanks for your detailed suggestion. It is corrected in the revised manuscript. Please see the Line: 521.

32. L509: Not sure if "disparate" is used correctly.

Response: Thanks for your detailed suggestion. It is changed to "different" in the revised manuscript. Please see the Line: 526.

33. L537-541: There is also the effect of cloud reductions, which "unmasks" some of the clear-sky component. Thus the masking effect may explain at least some of the clear-sky increasing trends over the SH midlatitudes. This may also help the following two paragraphs' analysis in contribution rates' trends.

Response: Thank you very much for your constructive comments that made the trend analysis more in-depth and complete. The discussion is added in the revised manuscript: "In addition, the reduction in clouds may also contribute to the increase in clear-sky atmospheric component. This is because cloud cover may mask some reflection of clear-sky components such as aerosols below the clouds (Qu and Hall, 2005; Donohoe and Battisti, 2011; Voigt et al., 2014; Stephens et al., 2015). Naturally, the decrease in cloud cover may reveal a portion of clear-sky atmospheric component"; "The opposite trend between the cloud component contribution and the clear-sky component contribution to some extent reflects the masking effect of clouds on clear-sky reflection. The reduction in clouds allows for a greater unmasking of the clear-sky component". Please see the Line: 559-563 and 616-618.

34. L583-584: This reflects previous results as well, such as Sledd and L'ecuyer (2021a) and Sledd and L'ecuyer (2021b).

Response: Thanks for your detailed suggestion. The relevant references are added in the revised manuscript. Please see the Line: 606-607.

35. Figure 5a: It may be good to label the "OBS" origin in the plot as "CERES" instead, since there are multiple observational data sets here, and it could be good to not regard CERES as absolute truth.

Response: Thank you very much for your detailed suggestion. It is corrected in the revised manuscript. Please see the Figure R6 (Figure 5a in the revised manuscript) and Line 625.

[Figure]

**Figure R6 (same as the Fig. 5a in the revised manuscript): (a) CCHZ-DISO system with 3-dimension for hemispheric difference of annual-average total RSR between NH and SH. The coordinate axis consists of three statistical indicators, normalized correlation coefficient (NCC) for x-axis, normalized absolute error (NAE) for y-axis, and normalized root mean square error (NRMSE) for z-axis. The DISO value is defined as the Euclidean distance between the three statistical indicators of each dataset and that of CERES (see Section 2.2.4 for details).**

36. L608-609: These sentences should be written in complete form.

    **Response:** Thanks for your detailed suggestion. It is corrected in the revised manuscript: It is corrected in the revised manuscript: "The blue, orange, yellow, purple and green bars indicate the statistical results for CERES EBAF, Cloud_cci AVHRR, ISCCP, MERRA-2, and ERA5, respectively". Please see the Line: 634-635.

37. L609: It could help to make the figure stand alone better if DISO is defined in the caption.

    **Response:** Thanks for your detailed suggestion. In the revised manuscript, the definition of DISO is added in the caption: "The DISO value is defined as the Euclidean distance between the three statistical indicators of each dataset and that of CERES (see Section 2.2.4 for details)". Please see the Line: 629-630.

38. L621: Suggest "Note that the good performance of dataset" → "Note that a data set may perform well because …"

    **Response:** Thanks for your detailed suggestion. It is corrected in the revised manuscript. Please see the Line: 647-648.

39. L654-655: I understand what this sentence is saying, but the grammar makes it difficult to read.

**Response:** Thanks for your detailed suggestion. It is corrected in the revised manuscript: "This bias of clear-sky atmospheric component between AVHRR and CERES are partly due to the fact that the current version of AVHRR dataset assumes a fixed aerosol optical thickness (AOD) of 0.05. This assumption will underestimate the AOD under conditions of high aerosol loading, resulting in a bias in the radiative flux (Stengel et al., 2020)." Please see the Line: 679-682.

40. L681: What is it that it exhibits the poorest of?

**Response:** We are sorry that our expression has caused confusion to the reviewer. In the revised manuscript, we correct the inappropriate statements: "Among all the datasets, MERRA-2 has the largest DISO value relative to the hemispheric difference in RSR observed by CERES, implying that it performing worst (Fig. 5a)." Please see the Line: 708-710.

41. L774: "Length of year" is an odd choice of words as it sounds like the year length is varying. It would be better to say "Length of time series [years]" instead.

**Response:** Thanks for your detailed suggestion. We have modified this statement both in the text and in the figure in the revised manuscript. Please see the Line 801, 814 and Figure 7.

[Figure]

**Figure R7 (same as the Fig. 7 in the revised manuscript): Cumulative annual mean for hemispheric differences of RSR and its components for (a) CERES, (b) AVHRR, (c) ISCCP, (d) MERRA-2 and (e)**

ERA5. That is, when Length of time series (years) = N, the hemispheric differences (NH-SH) of annual mean RSR are calculated from 2001 to 2000+N.The range of N varies due to the different record lengths of the datasets, with 1≤N≤21 for CERES,MERRA-2 and ERA5, while for AVHRR and ISCCP, 1≤N≤15.The black colour indicates the hemispheric difference of the total RSR, while the blue, green, and red colours correspond to the hemispheric differences of the three components, respectively (as y-axis labels in a).The shaded areas are the uncertainties of hemispheric difference of RSR for the given dataset. If the solid black line is within the shaded area, it indicates that the hemispheric symmetry in total RSR is credible within the uncertainty.

42. L782: Citation needs to be fixed as in L465.

Response: Thanks for your suggestion. We have fixed this incorrect citation in the revised manuscript. Please see the Line: 809.

43. L794: The quantification of the symmetry has been discussed in various ways throughout the literature; please clarify what it is that has not been shown before.

Response: We are sorry that our expression has caused confusion to the reviewer. Because our study and previous research have shown that CERES observes nearly equal reflection of solar radiation at the top of the atmosphere in both hemispheres. However, there is no fixed quantifiable measure to define this "nearly equal" condition. Therefore, in our paper, we discuss various numerical or criteria for assessing the symmetry of reflected solar radiation between hemispheres. We have modified this statement in the revised manuscript: "Although previous research, including our own study, have demonstrated that CERES observes nearly equal RSR at the TOA in both hemispheres, there is no fixed quantifiable measure to define this "nearly equal" condition. Therefore, we try to discuss various numerical or criteria for assessing the symmetry of RSR between hemispheres here". Please see the Line: 821-824.

44. L806: "More than two years" is a very imprecise description, please be specific.

Response: Thanks for your detailed suggestion. We have modified this statement in the revised manuscript: "AVHRR achieves it on scale of two years". Please see the Line: 835.

45. L812: What does "regional average" mean here? Across which spatial scale?

Response: We are sorry that our expression has caused confusion to the reviewer. The term "regional" here refers to a 1° x 1° grid area. It is clarified in the revised manuscript. Please see the Line: 842.

46. Figure 8d: There is a clear meridionally dependent bias with too much clear-sky atmospheric component in the tropics and too little at the poles in AVHRR. Is there an explanation for this or

can the authors comment on this? Perhaps water vapor affecting transmissivity can play a role here, since the t term in the transfer model can be reduced by increasing absorption of water vapor.

**Response:** Thank you very much for your comments. In the equatorial Pacific, we select a grid (179.5°E, 0.5°S) with a clear-sky atmospheric component showing a positive trend and plot the time series of the clear-sky atmospheric component from 1995-2015. We find that the value of clear-sky atmospheric component shows a steady downward trend during 1995-2000 and jumps at March 2001 (see Figure R8a). This jump coincides with the satellite transitions in the AVHRR_PM satellite platform: from NOAA14 to NOAA16 in April 2001. To identify the origins of sudden shifts in the radiative transfer process, we further investigate the radiative fluxes both at the top of the atmosphere and the surface. Under clear-sky conditions, the downward shortwave radiation fluxes at the top of the atmosphere (TOA) and the surface behave very stably (not shown), but the upward shortwave radiation flux at the TOA shows consistent changes with the clear-sky atmospheric component, while shortwave radiation flux at the surface show opposite change with the clear-sky atmospheric component (Fig. R8b, 8c). In addition, the temperature and water vapor atmospheric profiles used in the AVHRR radiative transfer model are obtained from ERA-Interim (Stengel et al., 2020), which may be relatively stable. Therefore, we speculate that the issue may not stem from the water vapor, but the surface and atmospheric reflection processes. Figure R9 shows the global distribution of surface upwelling shortwave radiation flux assuming clear-sky in March 2001 and the climatological anomalies for March 2001 relative to March 2002-2015. Such data jumps can be observed at low and mid-latitudes, especially in the tropics. The anomalies in the surface upwelling shortwave flux may be related to the orbital drift of NOAA14, which has resulted in a delay in the satellite's observation time and led to an increase in the measurement of solar zenith angle (SZA) (please see the comparison of Figure R10a and R10b) (Ji and Brown, 2017). The change in observation time for NOAA14 can be as much as 2 hours, with a delay from the original 2:00 PM transit to 4:00 PM. Furthermore, the documentation "ESA Cloud_cci Report on orbital drift correction for AVHRR" also mentions: "The direct impact of orbital drift of satellites is the gradual change in the time of observation. Change in observation time also implies change in the illumination conditions, i.e. solar zenith angles." Moreover, the surface albedo in this dataset is characterised by the bidirectional reflectance distribution function (BRDF), with different algorithms applied to the ocean and land. The SZA is a key parameter in the ocean surface albedo

calculation. On the one hand, under the influence of the Fresnel effect, the larger the SZA, the more energy is reflected from the ocean surface (Fig. R8c). On the other hand, the solar radiation reflected by the Earth's surface may be attenuated due to the increase in the total path length through the atmosphere with increased SZA, resulting in less reflected solar radiation reaching the top of the atmosphere (Fig. R8b). When the use of NOAA16 brings the observation time back to the initial time, the SZA also decreases, leading to a decrease in the jump of surface-reflected solar radiation and an increase in the jump of TOA reflected solar radiation. Furthermore, the variations in SZA resulting from orbital drift are most pronounced at lower latitudes. This is due to the curvature of the Earth, which leads to smaller daily SZA ranges at higher latitudes and consequently smaller rates of change over time. (Privette et al., 1995).

The detailed discussion is added in the revised manuscript: "The AVHRR clear-sky atmospheric component shows a clear positive trend over the oceans within 40° of latitude in both hemispheres, which is not seen in other datasets. Analysis of the 1995-2015 time series of the clear-sky atmospheric component (Fig. S14 a) for the grid over the equatorial Pacific Ocean (179.5°E, 0.5°S) shows a decreasing trend in the clear-sky atmospheric component during the period 1995-2000, with a jump in March 2001, coinciding with the transition from NOAA14 to NOAA16. The jump in data triggers spurious trends, which are also evident in the upwelling shortwave radiative fluxes at the TOA and surface, respectively (Fig. S14 b, c). The changes in the March 2001 surface and TOA upwelling shortwave radiation fluxes may be attributed to an increase in the solar zenith angle (SZA) due to satellite orbital drift (Ji and Brown, 2017). The SZA is a key parameter in the surface bidirectional reflectance distribution function (BRDF) used by AVHRR_PM_V3, which affects the surface reflected energy. Increased SZA may trigger the Fresnel effect and an increase in the atmospheric path length, respectively, hence affecting the reflected energy. Note that orbital drift-induced changes in SZA are most pronounced at low latitudes (Privette et al. 1995), which may explain the latitudinal dependence of these spurious trends." Please see the Line: 877-890.

[Figure]

**Figure R8 (same as the Fig. S14 in the revised manuscript): Monthly mean time series for clear-sky (a) the TOA atmospheric component, (b) the TOA upwelling shortwave flux and (c) the surface upwelling shortwave flux for a grid in the equatorial Pacific (179.5°E, 0.5°S) in AVHRR.**

[Figure]

**Figure R9: (a) Global distribution of clear-sky upwelling shortwave radiation flux at the surface in March 2001 in AVHRR. (b) Anomaly distribution of surface upwelling shortwave radiation flux in March 2001 relative to the 2002-2015 March climatology in AVHRR.**

[Figure]

**Figure R10: Solar zenith angle (top row) and sensor zenith angle (bottom row) for daytime in AVHRR. (a) and (c) are for 1 March 2001 (late in NOAA14), (b) and (d) are for 1 March 2002 (early in NOAA16).**

47. L875: "Unreal" is a loaded word to be used in scientific writing. Here and elsewhere I would suggest using phrases such as "not seen in CERES" rather than unreal or spurious.

    **Response:** Thanks for your detailed suggestion. We agree with you and have modified the expression in the revised manuscript. Please see the Line: 918.

48. L877: Clouds are not parameterized in observational data sets or radiative transfer models, or at least not in the same way that they are in reanalysis forecast models. I would suggest clarifying this earlier in the manuscript.

    **Response:** Thanks for your detailed suggestion. We correct the inappropriate statements in the revised manuscript: "This indicates that the cloud retrieval algorithms for satellite-based datasets, as well as the cloud parameterization schemes for reanalysis datasets, which are key sources of uncertainty in their cloud components, still require improvement." Please see the Line: 919-921.

49. L889: What is meant by "powerful"? Perhaps "robust"?

    **Response:** Thanks for your detailed suggestion. We have replaced "powerful" with "robust" in the revised manuscript. Please see the Line: 933.

50. L891-893: There have not been many compensation mechanisms suggested as of yet.

    **Response:** We agree with the reviewer. Indeed, the hypotheses for compensation mechanism that have been proposed so far are limited, including the movement of the ITCZ (Voigt et al., 2014) and the baroclinic activities in the mid-latitudes of the Southern Hemisphere (Hadas et al., 2023). In the

revised manuscript, we have corrected the inappropriate statements: "Several possible compensatory mechanisms have been proposed, which are not only limited by latitude but also have seasonal characteristics." Please see the Line: 935-937.

51. L908-910: This sentence makes it sound like the connection between the PDO and the North Atlantic clouds is trivial and well-documented, although I do not think this is what the authors meant to convey.

**Response:** We are sorry that our expression has caused confusion to the reviewer. We correct the inappropriate statements in the revised manuscript: "Decreasing trends in cloud component are mainly observed over the Northeast Pacific and North Atlantic near North America (Fig. S12c). The decreasing trend in cloud component over the Northeast Pacific may be associated with a shift in the Pacific Decadal Oscillation (PDO) phase from negative to positive, which leads to warmer SSTs in parts of the eastern Pacific, thus reducing low cloud cover and RSR (Loeb et al., 2018a; Loeb et al., 2020; Andersen et al., 2022). And the reduction in the North Atlantic cloud component may be related to a reduction in the optical thickness of low clouds due to a reduction in AOD (Park et al., 2024)"; "The decreasing trend in the cloud component is concentrated near the eastern Pacific and North Atlantic close to North America, which may be related to a decrease in low cloud cover and optical thickness, respectively. Specifically, the decrease in low cloud cover in the eastern Pacific is attributed to the increase in SST, which may be related to the shift of the PDO phase from negative to positive". Please see Line: 529-536 and 951-955.

52. L917-919: What is meant by the "the contribution rate from 30-40° N to the hemisphere"?

**Response:** We are sorry that our expression has caused confusion to the reviewer. It means "the rate of contribution of the reflected solar radiation in 30-40° N latitude zone to the hemispheric reflected solar radiation", which is shown in Figure 4d. It is clarified in the revised manuscript. Please see the Line: 962-964.

53. L921: The term "outstanding" is a bit too extreme here.

**Response:** Thanks for your detailed suggestion. We have replaced "outstanding" with "better" in the revised manuscript. Please see the Line: 966.

54. L955: There is a more recent IPCC Assessment Report that can be cited to better reflect the current state of understanding.

**Response:** Thanks for your detailed suggestion. We have cited the more recent IPCC Assessment Report (Forster et al., 2021) in the revised manuscript. Please see the Line: 999.

**Reference:**

Forster, P., Storelvmo, T., Armour, K., Collins, W., Dufresne, J.-L., Frame, D., Lunt, D., Mauritsen, T., Palmer, M., and Watanabe, M.: The Earth's energy budget, climate feedbacks, and climate sensitivity, 2021.

Hadas, O., Datseris, G., Blanco, J., Bony, S., Caballero, R., Stevens, B., and Kaspi, Y.: The role of baroclinic activity in controlling Earth's albedo in the present and future climates, Proceedings of the National Academy of Sciences, 120, e2208778120, https://doi.org/10.1073/pnas.2208778120, 2023.

Ji, L. and Brown, J. F.: Effect of NOAA satellite orbital drift on AVHRR-derived phenological metrics, International journal of applied earth observation and geoinformation, 62, 215-223, http://dx.doi.org/10.1016/j.jag.2017.06.013, 2017.

Loeb, N. G., Johnson, G. C., Thorsen, T. J., Lyman, J. M., Rose, F. G., and Kato, S.: Satellite and ocean data reveal marked increase in Earth's heating rate, Geophysical Research Letters, 48, e2021GL093047, https://doi.org/10.1029/2021GL093047, 2021.

Loeb, N. G., Mayer, M., Kato, S., Fasullo, J. T., Zuo, H., Senan, R., Lyman, J. M., Johnson, G. C., and Balmaseda, M.: Evaluating twenty‐year trends in Earth's energy flows from observations and reanalyses, Journal of Geophysical Research: Atmospheres, 127, e2022JD036686, https://doi.org/10.1029/2022JD036686, 2022.

Loeb, N. G., Doelling, D. R., Wang, H., Su, W., Nguyen, C., Corbett, J. G., Liang, L., Mitrescu, C., Rose, F. G., and Kato, S.: Clouds and the earth's radiant energy system (CERES) energy balanced and filled (EBAF) top-of-atmosphere (TOA) edition-4.0 data product, Journal of Climate, 31, 895-918, https://doi.org/10.1175/JCLI-D-17-0208.1, 2018.

Privette, J., Fowler, C., Wick, G., Baldwin, D., and Emery, W.: Effects of orbital drift on advanced very high resolution radiometer products: Normalized difference vegetation index and sea surface temperature, Remote Sensing of Environment, 53, 164-171, 1995.

Stengel, M., Stapelberg, S., Sus, O., Finkensieper, S., Würzler, B., Philipp, D., Hollmann, R., Poulsen, C., Christensen, M., and McGarragh, G.: Cloud_cci Advanced Very High Resolution Radiometer

post meridiem (AVHRR-PM) dataset version 3: 35-year climatology of global cloud and radiation properties, Earth System Science Data, 12, 41-60, https://doi.org/10.5194/essd-12-41-2020, 2020.

Trenberth, K. E. and Cheng, L.: A perspective on climate change from Earth's energy imbalance, Environmental Research: Climate, 1, 013001, 10.1088/2752-5295/ac6f74, 2022.

Trenberth, K. E. and Fasullo, J. T.: Tracking Earth's energy, Science, 328, 316-317, https://doi.org/10.1126/science.1187272, 2010.

Trenberth, K. E., Fasullo, J. T., and Balmaseda, M. A.: Earth's energy imbalance, Journal of Climate, 27, 3129-3144, https://doi.org/10.1175/JCLI-D-13-00294.1, 2014.

Voigt, A., Stevens, B., Bader, J., and Mauritsen, T.: Compensation of hemispheric albedo asymmetries by shifts of the ITCZ and tropical clouds, Journal of Climate, 27, 1029-1045, https://doi.org/10.1175/JCLI-D-13-00205.1, 2014.